# Adaptive Correction for Ensuring Conservation Laws in Neural Operators

## Abstract

Physical laws, such as the conversation of mass and momentum, are fundamental principles in many physical systems. Neural operators have achieved promising performance in learning the solutions to those systems, but often fail to ensure conservation. Existing methods typically enforce strict conservation via hand-crafted post-processing or architectural constraints, leading to limited model flexibility and adaptability. In this work, we propose a novel plug-and-play adaptive correction approach to ensure the conservation of fundamental linear and quadratic quantities for neural operator outputs. Our method introduces a lightweight learnable operator to adaptively enforce the target conservation law during training. This method allows the model to flexibly and adaptively correct its output to guarantee strict conservation. We provide a theoretical result showing that our correction method does not hamper the expression ability of neural operators and can potentially achieve lower reconstruction loss than their conservation-constrained counterparts. Our method is evaluated across multiple neural operator architectures and representative PDEs. Extensive experiments show that incorporating our correction method into baseline models significantly improves both accuracy and stability. In addition, the experimental results demonstrate that our approach consistently achieves superior performance over widely used conservation-enforcement techniques on various PDE benchmarks.

## 1. Introduction

Recent literature describes a number of neural network models that aim to solve partial differential equations (PDEs),

[1]Anonymous Institution, Anonymous City, Anonymous Region, Anonymous Country. Correspondence to: Anonymous Author <anon.email@domain.com>.

Preliminary work. Under review by the International Conference on Machine Learning (ICML). Do not distribute.

offering a flexible alternative to traditional numerical solvers (Azizzadenesheli et al., 2024; Farea et al., 2024). Traditional numerical methods rely on fine discretizations and hand-crafted meshes, which are computationally expensive, particularly in high-dimensional problems or those involving complex geometries and heterogeneous coefficients (Hughes, 2003; Johnson, 2009). In contrast, neural networks can learn solution operators directly from data, enabling mesh-free approximations and rapid inference across diverse inputs and resolutions (Faroughi et al., 2024; Liu et al., 2025; Lu et al., 2021). The data-driven paradigm has led to the development of a variety of neural network architectures and hybrid models (Hafiz et al., 2024). These approaches aim to capture both the physical structure and the dynamic behavior of PDE systems while enhancing scalability and generalization. Neural operators have been implemented through various architectures, including convolutional (Raonic et al., 2024; Ronneberger et al., 2015), Fourier-based (Li et al., 2021a), transformer-based (Cao, 2021), and graph-based networks (Sharma et al., 2024).

Despite their empirical success, standard data-driven neural operators lack inherent mechanisms to enforce physical conservation laws such as mass, momentum, or energy conservation. Conservation has long been a central research topic in traditional numerical methods, and many classical schemes are specifically designed to satisfy such laws (Hairer et al., 2006). In contrast, neural operators primarily focus on data approximation while often overlooking the underlying physical principles. This deficiency can lead to non-physical solutions, particularly in areas such as fluid dynamics, plasma physics, and wave propagation, where conservation is essential for physical fidelity and long-term stability. Violations of conservation laws not only reduce solution accuracy but also cause error accumulation over time, ultimately undermining the predictive reliability of long-time simulations. An empirical example of this behavior is shown in Figure 2.

**Limitations in recent conservation methods** To incorporate conservation laws into neural networks, previous work has proposed a variety of strategies, which can be broadly categorized into soft and hard constraint methods. Soft-constraint approaches typically add loss terms to penalize violations of the conservation laws, for example, by enforc-

*Table 1.* Comparison of different methods for enforcing conservation laws. Our approach is the first to simultaneously achieve **Strict**, **Plug-and-play**, **Learnable**, and **Nonlinear** properties.

| Method | Strict | Plug-and-play | Learnable | Nonlinear |
|---|---|---|---|---|
| Loss Constraint | ✗ | ✓ | ✗ | ✓ |
| Arch. Modification | ✓ | ✗ | ✓ | ✗ |
| Projection Method | ✓ | ✓ | ✗ | ✓ |
| **Ours** | ✓ | ✓ | ✓ | ✓ |

ing mass or energy conservation through integral loss functions (Li et al., 2021b; Wang et al., 2021; Wu et al., 2022). These methods can encourage physical consistency but cannot guarantee exact conservation, especially in long-term simulations. Ideally, constraint-enforcing methods should preserve physical laws while also enhancing model performance. In practice, however, soft-constraint approaches often improve conservation only at the expense of numerical accuracy and stability. Hard-constraint methods enforce conservation laws exactly, either by incorporating correction steps or post-processing mechanisms within the network pipeline (Cardoso-Bihlo & Bihlo, 2025; Geng et al., 2024), or by modifying the internal architecture to explicitly encode conservation properties (Liu et al., 2023b;a; Richter-Powell et al., 2022). While more rigorous in principle, post-processing-based methods typically rely on fixed, hand-crafted procedures that lack adaptability to diverse inputs. Architecture-based approaches are often restricted to enforcing linear conservation laws and may be incompatible with advanced or modular network architectures.

**The proposed adaptive correction method**  We propose an *adaptive correction* framework that enforces exact conservation laws in neural operators while preserving their flexibility and expressive power. Instead of relying on fixed, hand-crafted correction rules or restrictive architectural constraints, our method introduces a lightweight *learnable correction operator* that adaptively adjusts model outputs to strictly satisfy conservation laws. Our approach is distinguished by the following key properties:

- **Strict conservation**: The corrected outputs exactly satisfy the prescribed linear or quadratic conservation laws, eliminating long-term drift and ensuring physically consistent evolution.

- **Plug-and-play design**: The correction module can be seamlessly integrated into existing neural operators without modifying their original architectures or training pipelines.

- **Learnable and adaptive**: Unlike traditional projection or post-processing methods, the correction operator is learned from data, enabling it to adapt to diverse input distributions and dynamics.

- **Nonlinear compatibility**: The method supports both linear and quadratic conservation constraints, extending beyond the limitations of all architecture-based approaches.

## 2. Related work

Enforcing conservation laws in neural network-based PDE solvers has been an active research direction. One common approach is to augment the original data loss with an additional conservation loss term to enforce conservation constraints:

$$||u(\boldsymbol{x}, t) - u_{gt}(\boldsymbol{x}, t)||_{L^2} + \lambda ||\mathcal{G}(u)||_{L^2}, \quad (1)$$

where $u_{gt}(\boldsymbol{x}, t) : \Omega \times [0, T] \to \mathbb{R}^N$ and $u(\boldsymbol{x}, t) : \Omega \times [0, T] \to \mathbb{R}^N$ denote the ground-truth solution and the neural network approximation, respectively, and $\mathcal{G}(u) = 0$ encodes the conservation law to be enforced. This technique is widely used to encourage the approximate satisfaction of conservation laws (Chen & Qiao, 2025; De Ryck et al., 2024; Saharia et al., 2024; Wu et al., 2022). While such loss functions can improve the conservation behavior of models, they require careful tuning of the penalty weight $\lambda$ and cannot guarantee strict conservation. For physics-informed neural operators (Li et al., 2021b; Wang et al., 2021), their performance is highly sensitive to the choice of $\lambda$: even small variations can lead to significant degradation, as demonstrated in Section 4.3. This sensitivity substantially increases the difficulty of model tuning.

Exact conservation law preservation in neural networks for dynamical systems has recently been studied using various mathematical techniques. In Müller (2023), Noether's theorem is used to enforce Lie symmetries within neural networks. For Lagrangian systems, this approach guarantees the exact conservation of the associated first integrals. However, a key limitation of this method is that it applies only to Lagrangian systems, as non-Lagrangian systems fall outside the scope of Noether's theorem. A more general approach was proposed in (Cardoso-Bihlo & Bihlo, 2025), which incorporates projection-based corrections into the training process. At each step, the network output is adjusted by solving a constrained optimization problem to explicitly enforce conservation laws. Specifically, the corrected solution is obtained by solving

$$\min_{u} \ ||u(\boldsymbol{x}, t) - \hat{u}(\boldsymbol{x}, t)||_2$$
$$\text{s.t.} \quad \mathcal{G}(u) = 0, \quad (2)$$

where $\hat{u}$ denotes the output of a neural network and $\mathcal{G}(u) = 0$ encodes the conservation law. Related projection techniques have also been applied in the Fourier domain for Fourier Neural Operator (FNO) (Li et al., 2021a) to enforce conservation properties (Duruisseaux et al., 2024).

While these projection-based methods can guarantee exact conservation, they introduce substantial computational overhead, as a constrained optimization problem must be solved at every time step. Moreover, convergence of the projection procedure is often difficult to guarantee in practice, which limits scalability to large-scale or high-dimensional problems. Importantly, since the correction is not learned but imposed externally, these methods lack adaptivity and are more akin to post-processing steps.

Another line of work introduces conservation constraints via the divergence of skew-symmetric matrix-valued functions, effectively enforcing linear conservation laws, such as momentum conservation (Liu et al., 2023a; Richter-Powell et al., 2022). However, these methods are inherently limited to linear quantities and cannot generalize to nonlinear conservation laws such as norm or energy conservation.

Recently, a simple yet efficient post-processing method was proposed to enforce mass conservation in neural operator outputs (Geng et al., 2024). This constant adjustment approach computes the discrepancy between the predicted and initial total mass and corrects the prediction by applying a global offset. While this method is computationally efficient and guarantees exact mass conservation, it is inherently restricted to linear conserved quantities and does not extend to nonlinear constraints. Moreover, since the correction is applied externally rather than learned from data, it shares the same lack of adaptivity as projection-based methods. In this work, we propose a plug-and-play adaptive correction framework that overcomes these limitations by introducing a learnable operator to adjust neural operator outputs in a data-driven and input-dependent manner. Our method enforces conservation laws exactly while preserving and even enhancing predictive accuracy, thereby providing a more flexible, scalable, and physically consistent alternative to conventional correction techniques.

## 3. The Adaptive Correction Method

To ensure that neural operators respect fundamental conservation laws in physics, we introduce an adaptive correction method, which can handle two types of conservation laws in closed physical systems: linear and quadratic.

### 3.1. Correction for Linear Conservation Laws

Linear conservation laws, such as mass and momentum conservation, are characterized by the invariance of linear integral quantities over time. In mathematical terms, these laws require the following condition.

$$\frac{d}{dt} \int_{\Omega} u(\boldsymbol{x}, t)\, d\boldsymbol{x} = 0. \qquad (3)$$

Here, $u(\boldsymbol{x}, t)$ can represent a single physical quantity or the product of two quantities, for example, density for mass conservation, or density multiplied by velocity for momentum conservation. This equation implies the $\int_{\Omega} u(\boldsymbol{x}, t)\, d\boldsymbol{x}$ is constant over time, which we denote as $m_0$.

For simplicity, we consider the one-dimensional case. We first divide the spatial domain into $N$ equal small regions. Denote the $i$-th region as $\Delta x_i$. Suppose $U_i(t)$ is a discrete approximation of $\int_{\Delta x_i} u(\boldsymbol{x}, t)\, d\boldsymbol{x}$. Then discrete formulation of (3) can be characterized as follows.

$$\sum_{i=1}^{N} U_i = m_0. \qquad (4)$$

To ensure that (4) is satisfied for neural network output, we first propose a series of local correction operators as

$$\{\mathcal{L}_i\}_{i=1}^{N}, \quad \mathcal{L}_i : (m_0, \boldsymbol{U}) \to \boldsymbol{U}_{\text{new}},$$

where $\boldsymbol{U}$ is the discrete solution given by the neural operator and $\boldsymbol{U} = (U_1, U_2, \cdots, U_N)$. These operators are designed to map the original output $\boldsymbol{U}$ to a new output $\boldsymbol{U}_{\text{new}}$ that exactly satisfies (4). They are defined as follows

$$[\mathcal{L}_i(m_0, \boldsymbol{U})]_k = \begin{cases} U_k, & \text{if } k \neq i, \\ m_0 - \sum_{k \neq i} U_k & \text{if } k = i. \end{cases} \qquad (5)$$

Each operator $\mathcal{L}_i$ alters only the $i$-th entry of $\boldsymbol{U}$ to guarantee conservation, with all other entries unchanged.

In particular, when $\boldsymbol{U}$ represents multiple quantities, as in the case of momentum conservation laws, $\boldsymbol{U} = \boldsymbol{\rho} \cdot \boldsymbol{v}$, there exists an infinite number of solution pairs $(\boldsymbol{\rho}, \boldsymbol{v})$ to generate $\boldsymbol{U}_{\text{new}}$. One valid solution can be constructed, for example, by modifying only a single quantity at the $i$-th entry while keeping all other quantities fixed.

Relying solely on one local correction operator sacrifices flexibility at that point, as the value at that position is merely determined by the others. To overcome this, we combine all local correction operators to obtain a global correction operator. Specifically, we define the global mass-conserving operator as

$$\mathcal{L}(m_0, \boldsymbol{U}) = \sum_{i=1}^{N} \alpha_i \mathcal{L}_i(m_0, \boldsymbol{U}), \qquad (6)$$

where the coefficients $\alpha_i$ satisfy the constraint:

$$\sum_{i=1}^{N} \alpha_i = 1. \qquad (7)$$

With this constraint, the updated output $\boldsymbol{U}_{\text{new}}$ obtained by $\mathcal{L}(m_0, \boldsymbol{U})$ also satisfies (4). By parameterizing $\alpha_i$ with a

softmax function, the constraint is naturally satisfied while simultaneously enabling our global correction operator to be learnable.

Moreover, by substituting (5) into (6), we can simplify the expression and obtain

$$\mathcal{L}(m_0, \boldsymbol{U}) = \boldsymbol{U} + (m_0 - M(\boldsymbol{U}))\boldsymbol{A} \tag{8}$$

where $M(\boldsymbol{U}) = \sum_{i=1}^{N} U_i$ and $\boldsymbol{A}(k) = \alpha_k$. This formulation reveals that the global correction operator admits a straightforward implementation using the original output $\boldsymbol{U}$ and a learnable vector $\boldsymbol{A}$.

### 3.2. Correction for Quadratic Conservation Laws

Quadratic conservation laws, such as energy or norm conservation, require the invariance of quadratic quantities. A canonical example is the conservation of the squared norm:

$$\frac{d}{dt} \int_{\Omega} |u(\boldsymbol{x}, t)|^2 \, d\boldsymbol{x} = 0. \tag{9}$$

This equation implies that the quantity $\int_{\Omega} |u(\boldsymbol{x}, t)|^2 \, d\boldsymbol{x}$ remains constant. Let this constant be denoted as $c_0$. The corresponding 1D discrete form is given by

$$\sum_{i=1}^{N} U_i^2 = c_0. \tag{10}$$

Unlike the linear case, applying a series of conserved operators does not guarantee a conserved output for quadratic laws. Inspired by the form of (8) in the linear case, we introduce a quadratic correction operator $\mathcal{L}^q : (c_0, \boldsymbol{U}) \rightarrow \boldsymbol{U}_{\text{new}}$ by assuming $\boldsymbol{U}_{\text{new}}$ is a combination of the neural operator output $\boldsymbol{U}$ with a learnable vector $\boldsymbol{A}$:

$$\boldsymbol{U}_{\text{new}} = \lambda_1 \boldsymbol{U} + \lambda_2 \boldsymbol{A}. \tag{11}$$

To ensure that (10) is satisfied, the following condition must be met:

$$\sum_{i}^{N} (\lambda_1 U_i + \lambda_2 A_i)^2 = c_0. \tag{12}$$

If we define the following quantities

$$S_{U^2} = \sum_{i}^{N} U_i^2, \quad S_{A^2} = \sum_{i}^{N} A_i^2, \quad S_{UA} = \sum_{i}^{N} U_i A_i. \tag{13}$$

(12) then becomes

$$\lambda_1^2 S_{U^2} + 2\lambda_1 \lambda_2 S_{UA} + \lambda_2^2 S_{A^2} = c_0. \tag{14}$$

A real-valued solution for $\lambda_2$ exists if and only if

$$(2\lambda_1 S_{UA})^2 - 4S_{A^2}(\lambda_1^2 S_{U^2} - c_0) \geq 0. \tag{15}$$

Note that when $\lambda_1^2 S_{U^2} - c_0 \leq 0$, (15) always holds. In this case, (14) admits a real solution for $\lambda_2$, which can be obtained in closed form:

$$\lambda_2 = \frac{-\lambda_1 S_{UA} \pm \sqrt{(\lambda_1 S_{UA})^2 - S_{A^2}(\lambda_1^2 S_{U^2} - c_0)}}{S_{A^2}}. \tag{16}$$

To simplify the solution and ensure guaranteed feasibility, we assume $\lambda_1^2 S_{U^2} - c_0 = 0$, then we have,

$$\lambda_1 = \pm\sqrt{\frac{c_0}{S_{U^2}}}, \quad \lambda_2 = \mp\frac{2S_{UA}}{S_{A^2}}\sqrt{\frac{c_0}{S_{U^2}}} \tag{17}$$

Thus, we define the following operator $\mathcal{L}^q : (c_0, \boldsymbol{U}) \rightarrow \boldsymbol{U}_{\text{new}}$ for quadratic conservation laws.

$$\mathcal{L}^q(c_0, \boldsymbol{U}) = \sqrt{\frac{c_0}{S_{U^2} + \epsilon}}\boldsymbol{U} - \frac{2S_{UA}}{S_{A^2} + \epsilon}\sqrt{\frac{c_0}{S_{U^2} + \epsilon}}\boldsymbol{A}. \tag{18}$$

Here $\epsilon > 0$ is a small constant added for numerical stability, preventing small divisor problems when either $S_{U^2}$ or $S_{A^2}$ approach zero.

**Summary and Implementation Notes**  Together, the (8) and (18) define correction operators for linear and quadratic conservation laws respectively. In both cases, learnable coefficients $\boldsymbol{A}$ are introduced to endow the correction mechanism with adaptability and learning capability. In practice, the learnable coefficients $\boldsymbol{A}$ can be implemented as a set of trainable parameters or generated dynamically by a lightweight neural network such as a convolutional layer or a multilayer perceptron (MLP), conditioned on the type of the neural operator. For CNN-based neural operator like UNet, we choose convolutional layers to generate $\boldsymbol{A}$, while in the case of the Fourier Neural Operator (FNO) (Li et al., 2021a), which is resolution-invariant, we adopt an entry-wise MLP to generate $\boldsymbol{A}$ based on the output, which ensures that the correction mechanism inherits the resolution-invariance property of the FNO, thereby preserving their key advantages.

### 3.3. Theoretical Analysis

Imposing exact constraints on neural networks can restrict their expressive capacity and hinder data fitting in practice. The following theorem shows that our method preserves the expressive power of the neural operator, in the sense that it does not compromise its ability to fit data that satisfy the underlying conservation laws.

**Theorem 3.1.** *Define the following loss functions:*

$$L_1(u, u_{gt}) = \|u - u_{gt}\|,$$

$$L_2(u) = \begin{cases} \infty, & \mathcal{G}(u) \neq 0, \\ 0, & \mathcal{G}(u) = 0. \end{cases} \quad (19)$$

*Let $\mathcal{N}_F^\theta$ be the original neural operator, and $\mathcal{N}_A^\theta$ be the neural operator with our proposed adaptive correction. Define:*

$$\mathcal{N}_F^* = \arg\min_{\mathcal{N}_F^\theta} L_1(\mathcal{N}_F^\theta(u_0), u_{gt}) + L_2(\mathcal{N}_F^\theta(u_0)),$$

$$\mathcal{N}_A^* = \arg\min_{\mathcal{N}_A^\theta} L_1(\mathcal{N}_A^\theta(u_0), u_{gt}).$$

*We have*

$$L_1(\mathcal{N}_A^*(u_0), u_{gt}) \leq L_1(\mathcal{N}_F^*(u_0), u_{gt}). \quad (20)$$

**Proof.** Suppose that $\mathcal{N}_F^*$ exists (otherwise $\mathcal{N}_F^* = \emptyset$, (??) holds trivially). Then it must satisfy

$$L_2(\mathcal{N}_F^*(u)) = 0,$$

which implies

$$\mathcal{G}(\mathcal{N}_F^*(u)) = 0.$$

*Case 1: Linear conservation law.* Recall that for linear conservation laws,

$$\mathcal{G}(u) = m_0 - M(u).$$

By definition of the linear correction operator $\mathcal{L}$ in (8), if $\mathcal{G}(\mathcal{N}_F(u)) = 0$, then $\mathcal{L}$ leaves $\mathcal{N}_F(u)$ unchanged:

$$\mathcal{L}(\mathcal{N}_F^\theta(u)) = \mathcal{N}_F^\theta(u).$$

which implies

$$\mathcal{N}_A^\theta = \mathcal{L}(\mathcal{N}_F^\theta) = \mathcal{N}_F^\theta \text{ if } \mathcal{G}(\mathcal{N}_F^\theta(u)) = 0.$$

Therefore, we have

$$\begin{aligned} L_1(\mathcal{N}_A^*(u), u_{gt}) &= \min_{\mathcal{N}_A^\theta} L_1(\mathcal{N}_A^\theta(u), u_{gt}) \\ &\leq \min_{\{\mathcal{N}_A^\theta | \mathcal{G}(\mathcal{N}_F^\theta(u))=0\}} L_1(\mathcal{N}_A^\theta(u), u_{gt}) \\ &= \min_{\{\mathcal{N}_F^\theta | \mathcal{G}(\mathcal{N}_F^\theta(u))=0\}} L_1(\mathcal{N}_F^\theta(u), u_{gt}) \\ &= L_1(\mathcal{N}_F^*(u), u_{gt}). \end{aligned}$$

*Case 2: Quadratic conservation law.* For quadratic conservation laws, if $\mathcal{G}(u) = 0$, then the scaling factor satisfies

$$\frac{c_0}{S_u^2} = 1.$$

Thus, the quadratic correction operator $L^q$ defined in (18) maps

$$\mathcal{N}_F^*(u) \to \mathcal{N}_F^*(u) - \frac{2S_{uA}}{S_{A^2} + \epsilon}\boldsymbol{A},$$

Note that for any $u$, $\mathcal{L}$ can also leave $\mathcal{N}_F^\theta(u)$ unchanged:

$$\mathcal{N}_A^\theta(u) = \mathcal{L}(\mathcal{N}_F^\theta(u)) = \mathcal{N}_F^\theta(u) \quad \text{if } \boldsymbol{A} = 0.$$

By similar reasoning as in the linear case, we get

$$\begin{aligned} L_1(\mathcal{N}_A^*(u), u_{gt}) &= \min_{\mathcal{N}_A^\theta} L_1(\mathcal{N}_A^\theta(u), u_{gt}) \\ &\leq \min_{\{\mathcal{N}_A^\theta | \boldsymbol{A}=0\}} L_1(\mathcal{N}_A^\theta(u), u_{gt}) \\ &\leq \min_{\{\mathcal{N}_F^\theta | \mathcal{G}(\mathcal{N}_F^\theta(u))=0\}} L_1(\mathcal{N}_F^\theta(u), u_{gt}) \\ &= L_1(\mathcal{N}_F^*(u), u_{gt}). \end{aligned}$$

This completes the proof.

*Remark* 3.2. This theoretical result also carries practical significance. In practice, directly enforcing conservation by optimizing the objective $\mathcal{L}_1 + \lambda\|\mathcal{G}(u)\|_{L^2}$ with a large penalty parameter $\lambda$ often results in unstable or inefficient training. Theorem 3.1 indicates that, in the limiting case $\lambda \to \infty$, such constrained optimization can be effectively realized by training a neural operator equipped with our adaptive correction, without explicitly introducing a stiff penalty term, thereby leading to more stable and efficient training.

## 4. Experiments

We now conduct extensive experiments to evaluate the effectiveness of the proposed adaptive correction method in enforcing mass and norm conservation across a variety of neural operator architectures and benchmark PDEs. Specifically, we evaluate our method on three representative neural architectures: UNet (Ronneberger et al., 2015), the Galerkin Transformer Neural Operator (GTNO) (Cao, 2021), and the Fourier Neural Operator (FNO) (Li et al., 2021a). Our implementation of the GTNO and the FNO builds upon the publicly available codebases at https://github.com/scaomath/galerkin-transformer and https://github.com/neuraloperator/neuraloperator, respectively. More implementation details are provided in Appendix A. The learnable coefficients $\boldsymbol{A}$ are parameterized by a single convolutional layer for UNet and GTNO, and by a lightweight MLP with three hidden layers for FNO, and this setup is used consistently across all the experiments. The results demonstrate that our method not only strictly preserves the targeted conservation laws but also consistently improves predictive accuracy over the original models. We further compare our method with loss-based correction and projection-based correction on FNO to highlight its advantages.

### 4.1. Benchmark PDEs

We select three mass-conserving and three norm-conserving equations to evaluate the effectiveness of the proposed adaptive correction method on each type of conservation law. The selected equations are listed below and the data generation details are provided in Appendix B.

**Mass Conservation Equations**

- **Transport Equation (TE):**

$$u_t + \nabla \cdot (u\boldsymbol{v}) = 0, \quad \boldsymbol{x} \in \Omega, \quad t > 0, \qquad (21)$$

where $u = u(\boldsymbol{x},t)$ represents the scalar field, $\boldsymbol{v} = \boldsymbol{v}(\boldsymbol{x},t)$ is the velocity field, $\boldsymbol{x} \in \mathbb{R}^d$ denotes the spatial coordinates, and $t$ is time. This equation describes the advection of a scalar field, and mass conservation is satisfied when the system is closed, or the boundary condition is periodic. We test the 2D transport equation with $\boldsymbol{v}(\boldsymbol{x},t) \equiv (1,1)$ on $\Omega = [0,1]^2$ with the periodic boundary condition. The operator we aim to learn is the mapping between the initial condition $u(\boldsymbol{x},0)$ to $u(\boldsymbol{x},\Delta t)$ with $\Delta t = 0.05$.

- **Conservative Allen-Cahn Equation (CAC):**

$$u_t = \nabla \cdot (\epsilon \nabla u) + u - u^3$$
$$- \frac{1}{|\Omega|} \int_\Omega (u - u^3)\, d\boldsymbol{x}, \qquad \boldsymbol{x} \in \Omega,\ t > 0. \tag{22}$$

where $u = u(\boldsymbol{x},t)$ is the scalar field, $\epsilon > 0$ is a small constant related to interface thickness. This phase-field model is used in material science, and the conservation of mass is critical for accurate simulations. We test the 2D Allen-Cahn Conservative Equation with $\epsilon = 0.01$ on $\Omega = [0,1]^2$ with a periodic boundary condition. The mapping to be learned is $u(\boldsymbol{x},0) \rightarrow u(\boldsymbol{x},\Delta t)$ with $\Delta t = 0.5$.

- **Shallow Water Equations (SWE):**

$$\begin{cases} h_t + \nabla \cdot (h\boldsymbol{u}) = 0, \\ (h\boldsymbol{u})_t + \nabla \cdot \left(h\boldsymbol{u} \otimes \boldsymbol{u} + \frac{1}{2}gh^2\boldsymbol{I}\right) = 0, \end{cases} \boldsymbol{x} \in \Omega,\ t > 0, \tag{23}$$

Here, $h = h(\boldsymbol{x},t)$ is the fluid height, $\boldsymbol{u} = \boldsymbol{u}(\boldsymbol{x},t)$ is the velocity field, $g$ is the gravitational acceleration, and $\boldsymbol{I}$ denotes the identity matrix. This system models the dynamics of incompressible, inviscid shallow water flow. The first equation enforces mass conservation, which holds under periodic or no-flux boundary conditions. In our experiments, we consider the 2D shallow water equations with periodic boundary conditions. The dataset is obtained from the PDEBench dataset (Takamoto et al., 2022). We aim to learn the mapping $h(\boldsymbol{x},0) \rightarrow h(\boldsymbol{x},\Delta t)$ with $\Delta t = 0.01$.

- **Compressible Navier-Stokes Equation (CNS):**

$$\partial_t \rho + \nabla \cdot (\rho\mathbf{v}) = 0,$$
$$\rho(\partial_t \mathbf{v} + \mathbf{v} \cdot \nabla \mathbf{v}) = -\nabla p + \eta \Delta \mathbf{v} + \left(\zeta + \frac{\eta}{3}\right)\nabla(\nabla \cdot \mathbf{v}),$$
$$\partial_t \varepsilon + \nabla \cdot \left((\varepsilon + p + \frac{1}{2}\rho|\mathbf{v}|^2)\mathbf{v} - \mathbf{v} \cdot \boldsymbol{\tau}\right) = 0. \tag{24}$$

Here, $\rho$ denotes the fluid density, $\mathbf{v}$ the velocity field, $p$ the pressure, and $\varepsilon$ the internal energy density. The parameters $\eta$ and $\zeta$ represent the shear and bulk viscosity coefficients, respectively, and $\boldsymbol{\tau}$ denotes the viscous stress tensor. We test the 2D compressible Navier-Stokes equation with $\eta = 0.01$ and $\zeta = 0.01$ on periodic boundary condition. The mapping to be learned is $\boldsymbol{U}(\boldsymbol{x},0) \rightarrow \boldsymbol{U}(\boldsymbol{x},\Delta t)$ with $\Delta t = 0.1$ and $\boldsymbol{U} = (\boldsymbol{v}_x, \boldsymbol{v}_y, \rho, p)$.

**Norm Conservation Equations**

- **Transport Equation (TE):** As in the setting of the mass conservation case, the $L^2$ norm of $u$ is also conserved over time.

- **Schrödinger Equation:**

$$\text{(LSE)} \quad i\psi_t + \frac{1}{2}\Delta\psi + V(\boldsymbol{x})\psi = 0, \quad \boldsymbol{x} \in \Omega,\ t > 0,$$
$$\text{(NSE)} \quad i\psi_t + \frac{1}{2}\Delta\psi + \lambda|\psi|^2\psi = 0, \quad \boldsymbol{x} \in \Omega,\ t > 0. \tag{25}$$

where LSE and NSE stand for the linear Schrödinger equation and the nonlinear Schrödinger equation, respectively. Here, $\psi = \psi(\boldsymbol{x},t)$ is a complex-valued wave function, $i$ is the unit imaginary number and $\Delta$ denotes the Laplacian operator. A key property of both the linear and nonlinear Schrödinger equations is the conservation of the $L^2$ norm $\int_\Omega |\psi(\boldsymbol{x},t)|^2 d\boldsymbol{x}$, which is fundamental in quantum mechanics. In our experiments, $\Omega = [0,1]$ and we use $V(\boldsymbol{x}) \equiv 1$ for the linear case and set $\lambda = 1$ for the nonlinear case, with periodic boundary conditions in both. The mapping we aim to learn is $\psi(\boldsymbol{x},0) \rightarrow \psi(\boldsymbol{x},\Delta t)$ with $\Delta t = 0.025$.

### 4.2. Accuracy Tests for Adaptive Correction

We now evaluate the prediction accuracy of all of the baseline models, with results reported in Table 2. We find that all of the neural operators equipped with the proposed adaptive correction outperform their original counterparts on all of the test benchmarks. In addition to prediction accuracy, our method also improves the preservation of the inherent physical properties. Figure 1 shows the evolution of the $L^2$ norm of $\psi$, which describes mass density or probabilistic density, for both FNO and FNO with our adaptive correction. The standard FNO deviates from the ground truth and exhibits error growth over time, whereas the FNO with adaptive correction remains closely aligned with the

*Table 2.* Prediction error on test dataset for the original neural operators and their counterparts with the proposed adaptive correction method.

| Conservation | Equation | UNet | | GTNO | | FNO | |
|---|---|---|---|---|---|---|---|
| | | original | ours | original | ours | original | ours |
| Mass | TE | $1.08 \pm 0.14$e-1 | $\mathbf{0.83 \pm 0.09}$e-1 | $9.11 \pm 0.76$e-2 | $\mathbf{8.15 \pm 0.87}$e-2 | $8.29 \pm 0.12$e-2 | $\mathbf{8.04 \pm 0.11}$e-2 |
| | CAC | $1.48 \pm 0.08$e-2 | $\mathbf{1.42 \pm 0.14}$e-2 | $4.95 \pm 0.17$e-2 | $\mathbf{4.47 \pm 0.18}$e-2 | $2.01 \pm 0.26$e-2 | $\mathbf{1.65 \pm 0.19}$e-2 |
| | SWE | $2.78 \pm 0.29$e-3 | $\mathbf{1.40 \pm 0.08}$e-3 | $1.55 \pm 0.14$e-2 | $\mathbf{0.78 \pm 0.04}$e-2 | $2.57 \pm 0.09$e-3 | $\mathbf{2.32 \pm 0.14}$e-3 |
| | CNS | $6.72 \pm 0.04$e-1 | $\mathbf{4.59 \pm 0.06}$e-1 | $2.66 \pm 0.04$e-1 | $\mathbf{1.89 \pm 0.02}$e-1 | $1.50 \pm 0.10$e-1 | $\mathbf{1.30 \pm 0.03}$e-1 |
| Norm | TE | $1.08 \pm 0.14$e-1 | $\mathbf{0.82 \pm 0.05}$e-1 | $9.11 \pm 0.76$e-2 | $\mathbf{8.21 \pm 0.89}$e-2 | $8.29 \pm 0.12$e-2 | $\mathbf{8.01 \pm 0.16}$e-2 |
| | LSE | $1.84 \pm 0.07$e-1 | $\mathbf{1.10 \pm 0.17}$e-1 | $1.66 \pm 0.21$e-3 | $\mathbf{0.80 \pm 0.10}$e-3 | $3.77 \pm 0.28$e-3 | $\mathbf{3.22 \pm 0.09}$e-3 |
| | NSE | $2.40 \pm 0.03$e-1 | $\mathbf{2.33 \pm 0.12}$e-1 | $1.77 \pm 0.10$e-2 | $\mathbf{1.43 \pm 0.04}$e-2 | $3.82 \pm 1.06$e-2 | $\mathbf{3.02 \pm 0.51}$e-2 |

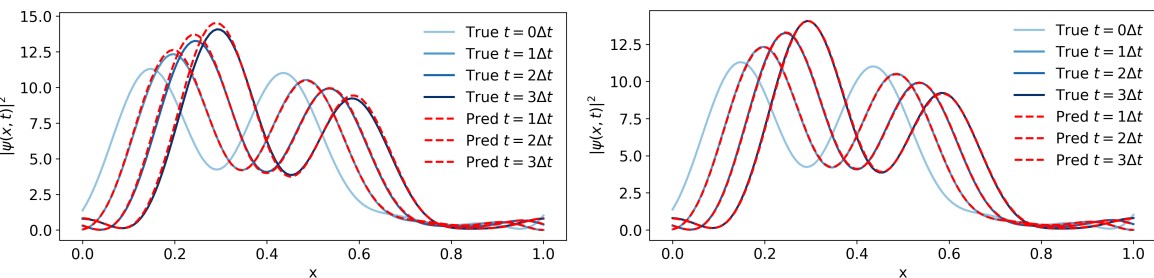

*Figure 1.* Solution dynamics of the linear Schrödinger equation obtained with the baseline FNO and our proposed method over time, starting from $t = 0$ (solid light blue line). $\Delta t$ denotes the prediction time interval. Left: FNO. Right: FNO with our method.

*Table 3.* Prediction error on test dataset for the original FNO and FNO with conservation methods.

| Conservation Laws | Equation | FNO | Loss | Projection | Ours |
|---|---|---|---|---|---|
| Mass | TE | $8.29 \pm 0.12$e-2 | $8.16 \pm 0.11$e-2 | $8.14 \pm 0.14$e-2 | $\mathbf{8.04 \pm 0.11}$e-2 |
| | CAC | $2.01 \pm 0.26$e-2 | $2.24 \pm 0.34$e-2 | $99.7 \pm 0.47$e-2 | $\mathbf{1.65 \pm 0.19}$e-2 |
| | SWE | $2.57 \pm 0.09$e-3 | $2.82 \pm 0.14$e-3 | $3.01 \pm 0.34$e-3 | $\mathbf{2.32 \pm 0.14}$e-3 |
| | CNS | $1.50 \pm 0.10$e-1 | $7.00 \pm 0.02$e-1 | $1.34 \pm 0.03$e-1 | $\mathbf{1.30 \pm 0.03}$e-1 |
| Norm | TE | $8.29 \pm 0.12$e-2 | $8.17 \pm 0.24$e-2 | $8.34 \pm 0.24$e-2 | $\mathbf{8.01 \pm 0.16}$e-2 |
| | LSE | $3.77 \pm 0.28$e-3 | $4.08 \pm 0.87$e-3 | $3.94 \pm 0.98$e-3 | $\mathbf{3.22 \pm 0.09}$e-3 |
| | NSE | $3.82 \pm 1.06$e-2 | $3.75 \pm 0.32$e-2 | $3.52 \pm 0.35$e-2 | $\mathbf{3.02 \pm 0.51}$e-2 |

*Table 4.* Conservation error on test dataset for the original FNO and FNO with conservation methods.

| Conservation Laws | Equation | FNO | Loss | Projection | Ours |
|---|---|---|---|---|---|
| Mass | TE | $6.42 \pm 1.2$ | $5.27 \pm 1.6$ | $\mathbf{0.00 \pm 0.0}$ | $\mathbf{0.00 \pm 0.0}$ |
| | CAC | $46.7 \pm 7.4$ | $41.7 \pm 5.2$ | $\mathbf{0.00 \pm 0.0}$ | $\mathbf{0.00 \pm 0.0}$ |
| | SWE | $13.3 \pm 1.4$ | $9.72 \pm 0.9$ | $\mathbf{0.00 \pm 0.0}$ | $\mathbf{0.00 \pm 0.0}$ |
| | CNS | $\geq 1000$ | $\geq 1000$ | $\mathbf{0.00 \pm 0.0}$ | $\mathbf{0.00 \pm 0.0}$ |
| Norm | TE | $31.6 \pm 5.4$ | $26.2 \pm 5.8$ | $\mathbf{0.00 \pm 0.0}$ | $\mathbf{0.00 \pm 0.0}$ |
| | LSE | $2.55 \pm 0.4$ | $2.27 \pm 0.5$ | $\mathbf{0.00 \pm 0.0}$ | $\mathbf{0.00 \pm 0.0}$ |
| | NSE | $13.5 \pm 6.2$ | $11.2 \pm 4.7$ | $\mathbf{0.00 \pm 0.0}$ | $\mathbf{0.00 \pm 0.0}$ |

ground truth even after multiple prediction steps. Figure 2 illustrates how prediction errors evolve over time: while the standard FNO quickly diverges from the ground truth, the corrected FNO remains closely aligned, demonstrating that our method enhances the stability of neural operators in long-term prediction. Additional visualizations for other equations are provided in Appendix C.

### 4.3. Adaptive Correction vs. Other Methods

In this subsection, we compare the proposed adaptive correction method with the loss-based method and the projection

method on the FNO model. The prediction error is calculated as the relative $L^2$ error between the predicted and true solutions, while the conservation error is calculated as the discrepancy between the sum of the conservative quantity at all grid points. All results are reported in Table 3 and Table 4, respectively.

**Adaptive Correction vs. Loss-based Method.** Loss-based approaches enforce conservation by augmenting the training objective with a penalty term on conservation violations. We implement this strategy by training FNO with the loss in Eq. 1 and evaluate its performance on the transport equation under different penalty weights $\lambda$. As shown in Table 5, the performance of the loss-based method is highly sensitive to the choice of $\lambda$: overly large values degrade prediction accuracy, while small perturbations of $\lambda$ can lead to abrupt performance changes, as evidenced by the fluctuation at $\lambda = 10^{-3}$ in the norm conservation case.

Using empirically optimal $\lambda$ values identified on the transport equation, we further evaluate the loss-based method on other PDE benchmarks. Results in Tables 3 and 4 show that these tuned hyperparameters do not generalize well across equations, with particularly poor performance on the compressible Navier–Stokes equation. This highlights a fundamental limitation of loss-based methods: their effectiveness relies on careful, problem-specific hyperparameter tuning.

In contrast, the proposed adaptive correction method requires no manual tuning and consistently outperforms both

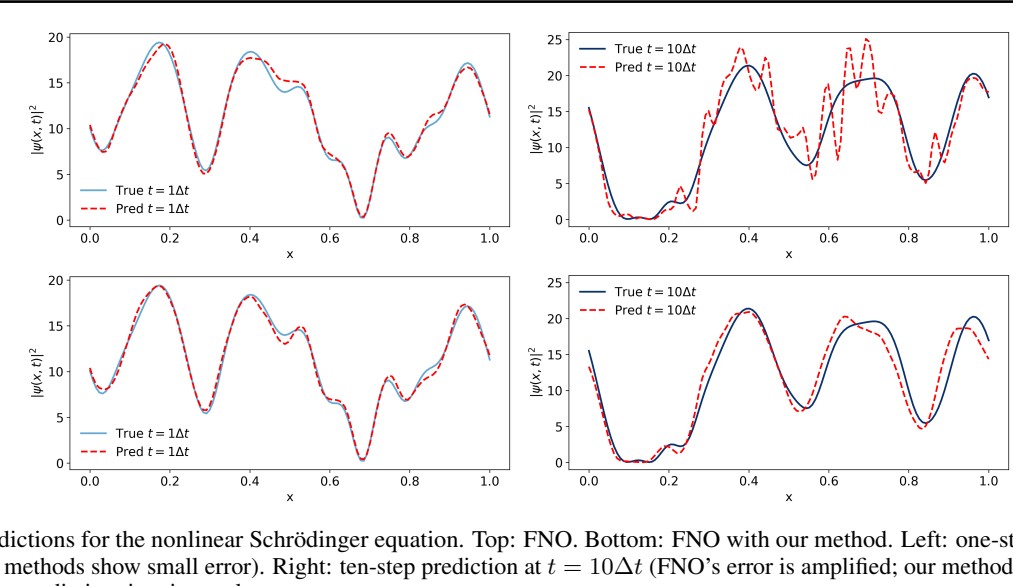

*Figure 2.* Predictions for the nonlinear Schrödinger equation. Top: FNO. Bottom: FNO with our method. Left: one-step prediction at $t = \Delta t$ (both methods show small error). Right: ten-step prediction at $t = 10\Delta t$ (FNO's error is amplified; our method remains stable). $\Delta t$ denotes the prediction time interval.

*Table 5.* Prediction error (%) for the FNO with different conservation loss and different $\lambda$.

| $\lambda$ | 0 | 1e-4 | 1e-3 | 1e-2 |
|---|---|---|---|---|
| Mass conservation | $8.29 \pm 0.12$ | $8.25 \pm 0.15$ | **8.16±0.11** | $9.29 \pm 0.12$ |

| $\lambda$ | 0 | 1e-5 | 1e-4 | 1e-3 |
|---|---|---|---|---|
| Norm conservation | $8.29 \pm 0.12$ | $8.35 \pm 0.21$ | **8.17±0.16** | $90.1 \pm 0.32$ |

*Table 6.* Prediction error (%) on test dataset for the original FNO, **FNO\*** (FNO with a learnable matrix appended) and FNO with the proposed adaptive correction method.

| Conservation Laws | Equation | FNO | FNO\* | Ours |
|---|---|---|---|---|
| Mass Conservation | TE | $8.29 \pm 0.12$ | $8.26 \pm 0.18$ | **8.04±0.11** |
| | CAC | $2.01 \pm 0.26$ | $2.23 \pm 0.85$ | **1.65±0.19** |
| | SWE | $0.26 \pm 0.01$ | $0.28 \pm 0.01$ | **0.23±0.01** |
| | CNS | $15.0 \pm 1.00$ | $15.2 \pm 0.58$ | **13.0±0.29** |
| Norm Conservation | TE | $8.29 \pm 0.12$ | $8.26 \pm 0.18$ | **8.01±0.16** |
| | LSE | $0.38 \pm 0.03$ | $1.61 \pm 0.34$ | **0.32±0.02** |
| | NSE | $3.82 \pm 1.06$ | $4.84 \pm 1.52$ | **3.02±0.51** |

the original FNO and the loss-based approach across all evaluated PDEs. Moreover, while the loss-based method only modestly reduces conservation errors, our method enforces the conservation laws up to machine precision, achieving exact satisfaction in the model outputs.

**Adaptive Correction vs. Projection method** The projection methods are implemented by solving (2) both in training and in prediction. As shown in Table 3 and Table 4, while the projection method enforces exact conservation, it does not reduce the prediction error and, in the case of the conservative Allen–Cahn equation, significantly increases it. In contrast, the adaptive correction method not only enforces conservation laws effectively but also maintains or improves predictive accuracy compared to both baselines. Notably, the conservation error with the adaptive correction method is consistently at machine precision, highlighting its ability to enforce physical constraints without sacrificing accuracy.

### 4.4. Ablation Study

We next conduct an ablation study to verify that the observed performance gains stem from the targeted enforcement of conservation laws rather than the mere addition of learnable parameters. To this end, we compare our approach with a baseline that appends the same MLP to the output of the original FNO architecture without any conservation-driven

correction. The results, summarized in Table 6, show that the adaptive correction method consistently outperforms the baseline.

## 5. Conclusion and Limitations

In this work, we propose an adaptive correction approach that dynamically adjusts the output of neural operators to satisfy conservation laws. We have conducted a comprehensive set of experiments on various neural operator architectures and PDEs, demonstrating the effectiveness of our method for mass and norm conservation. The results show that our approach not only exactly enforces the desired conservation laws but also improves the overall accuracy and stability of the prediction. Further comparisons with existing conservation techniques highlight the superiority of our adaptive correction method. At present, our approach focuses on enforcing a single conservation law at a time, and is limited to linear and quadratic forms. Extending the framework to simultaneously enforce multiple, and more general conservation laws, and to handle higher-order conservation laws represents a promising direction for future work.

## Impact Statement

This work addresses a fundamental limitation in neural operator modeling: the inability to rigorously enforce physical conservation laws. By introducing a lightweight, adaptive correction operator, our method allows neural operators to maintain physical consistency while improving reconstruction accuracy and stability. The approach is architecture-agnostic and can be applied to a wide range of PDEs, making it broadly useful for scientific machine learning. Beyond improving performance on standard benchmarks, this framework offers a principled way to integrate physical knowledge into data-driven models, potentially impacting areas such as fluid dynamics, climate modeling, and material science, where fundamental physical laws are critical.

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

# Appendix

## A. Implementations Details for FNO

### A.1. UNet configurations

The implemented UNet model is a five-level architecture with an initial depth of four encoding layers, followed by a bottleneck, and four decoding layers. The input channels are configurable (default: 3). Starting with 16 feature channels, the number doubles at each encoding step, reaching 256 channels at the bottleneck. Downsampling is performed using max-pooling layers with a kernel size and stride of two, while upsampling uses transposed convolutions with the same parameters. Each convolutional block consists of two convolutional layers with a kernel size of three, circular padding, Tanh activations, and no bias. Skip connections are used to concatenate encoder features with the corresponding decoder layers, enabling the network to integrate hierarchical and spatial details effectively. The output layer uses a $1 \times 1$ convolution to adjust the feature map dimensions to the desired number for outputs. The learning rate for UNet is 1e-4 across all experiments.

### A.2. GTNO configurations for different PDE benchmarks.

**GTNO configurations for transport, conservative Allen-Cahn, Shallow water equation and compressible Navier-Stokes equation.**

- **Learning rate:** $1 \times 10^{-4}$

- **Position dimension:** 2

- **Hidden dimension:** 128

- **Number of feature layers:** 0

- **Number of encoder layers:** 6

- **Number of attention heads:** 4

- **Feedforward dimension:** 256

- **Feature extraction type:** none

- **Attention type:** Galerkin

- **Xavier initialization scale:** 0.01

- **Diagonal weight:** 0.01

- **Symmetric initialization:** False

- **Layer normalization:** False

- **Attention normalization:** True

- **Normalization epsilon:** $1 \times 10^{-7}$

- **Batch normalization:** False

- **Return attention weights:** False

- **Return latent representations:** False

- **Decoder type:** IFFT2

- **Spatial dimension:** 2

- **Spatial fully connected layer:** True

- **Upsampling mode:** Interpolation

- **Downsampling mode:** Interpolation

- **Frequency dimension:** 32

- **Boundary condition:** Dirichlet

- **Number of regressor layers:** 2

- **Fourier modes:** 12

- **Regressor activation:** SiLU

- **Downscaler activation:** ReLU

- **Upscaler activation:** SiLU

- **Last activation:** True

- **Dropout rate:** 0.0

- **Downscaler dropout:** 0.05

- **Upscaler dropout:** 0.0

- **FFN dropout:** 0.05

- **Encoder dropout:** 0.05

- **Decoder dropout:** 0.0

- **Downscaler size:** 32

**GTNO configurations for Linear and Nonlinear Schrödinger Equations.**

- **Learning rate:** $1 \times 10^{-4}$

- **Position dimension:** 1

- **Hidden dimension:** 96

- **Number of feature layers:** 0

- **Number of encoder layers:** 4

- **Number of attention heads:** 1

- **Feedforward dimension:** 192

- **Feature extraction type:** none

- **Attention type:** Fourier

- **Xavier initialization scale:** 0.01

- **Diagonal weight:** 0.01

- **Symmetric initialization:** False

- **Layer normalization:** False

- **Attention normalization:** True

- **Normalization epsilon:** $1 \times 10^{-7}$

- **Batch normalization:** False

- **Return attention weights:** False

- **Return latent representations:** False

- **Decoder type:** IFFT

- **Spatial dimension:** 1

- **Spatial fully connected layer:** False

- **Upsampling mode:** Interpolation

- **Downsampling mode:** Interpolation

- **Frequency dimension:** 48

- **Boundary condition:** Dirichlet

- **Number of regressor layers:** 2

- **Fourier modes:** 16

- **Dropout rate:** 0.0

- **FFN dropout:** 0.0

- **Encoder dropout:** 0.0

- **Decoder dropout:** 0.0

### A.3. FNO configurations for different PDE benchmarks.

**FNO configurations for transport, conservative Allen-Cahn, shallow water equation and compressible Navier-Stokes equation.**

- **Learning rate:** $1.5 \times 10^{-3}$

- **Number of Fourier modes (height):** 48

- **Number of Fourier modes (width):** 48

- **Input channels:** 3 (6 for CNS)

- **Lifting channels:** 256

- **Hidden channels:** 64

- **Output channels:** 1 (4 for CNS)

- **Projection channels:** 256

- **Number of layers:** 4

- **Normalization:** Group normalization

- **Skip connection:** Linear

- **Use MLP:** True

- **Tensor factorization:** Tucker (null for CNS)

- **Rank:** 1

**FNO configurations for Linear and Nonlinear Schrödinger Equations.**

- **Learning rate:** $1.5 \times 10^{-3}$

- **Number of Fourier modes:** 32

- **Input channels:** 4

- **Lifting channels:** 128

- **Hidden channels:** 64

- **Output channels:** 2

- **Projection channels:** 128

- **Number of layers:** 4

- **Normalization:** Group normalization

- **Skip connection:** Linear

- **Use MLP:** True

- **Tensor factorization:** Tucker

- **Rank:** 1

### A.4. Implementation Details of the Adaptive Correction Module

For UNet and GTNO, $A$ is parameterized by a convolutional layer with kernel size 3 in all experiments, taking as input the concatenation of the original output and the final feature maps. For FNO, $A$ is parameterized by a lightweight three-layer MLP, whose hidden dimension is set to twice the number of output channels.

### A.5. Training details

For all experiments, we train the baselines with Adam optimizer and a learning rate decay of 0.5 is applied every 100 epochs. For the training of UNet and GTNO, we train 500 epochs for all equations. For the training of FNO models, we train 100 epochs for the transport equation and 500 epochs for all other equations.

The spatial resolution for 2D equations, including the transport equation, the conservative Allen–Cahn equation, and the shallow water equation, is set to $128 \times 128$. For 1D equations such as linear and nonlinear Schrödinger equations, the data are represented with shape $(2, 128)$ to accommodate complex-valued functions.

We used 2,000 training samples for the transport equation, and 1000 training samples for all other equations. Each experiment was evaluated on an additional 1,000 test samples. All experiments were conducted on a single NVIDIA A100 GPU with 80GB of memory. To ensure the reliability of our results, each experiment is repeated three times using different random seeds.

## B. Data Generation Details for PDEs

- **Transport Equation:**
$$u_t + \nabla \cdot (u\boldsymbol{v}) = 0, \quad x \in \Omega, \quad t > 0, \tag{26}$$

  For the 2D transport equation with constant velocity field $\boldsymbol{v} \equiv (1, 1)$, the analytical solution is given by:
$$u(x, y, t) = u_0(x - t, y - t, 0), \tag{27}$$

  where the initial condition is defined as:
$$u_0(x, y, t) = A \sin(2\pi k_1 x) \sin(2\pi k_2 y), \tag{28}$$

  with the amplitude $A$ sampled uniformly from $[2.5, 3]$ and the wave numbers $k_1, k_2$ randomly selected from $0, 1, 2, 3$.

- **Conservative Allen-Cahn Equation:**

$$u_t = \nabla \cdot (\epsilon \nabla u) + u - u^3 - \frac{1}{|\Omega|} \int_\Omega u - u^3 d\boldsymbol{x}, \quad \boldsymbol{x} \in \Omega, \quad t > 0, \tag{29}$$

We simulate the 2D conservative Allen–Cahn equation using the forward Euler method with a time step size of $10^{-5}$ and $\epsilon = 0.01$. Periodic boundary conditions are applied. The initial condition values are sampled independently and uniformly from the interval $[-1, 1]$ at each spatial point.

- **Shallow Water Equations:**

$$\begin{cases} h_t + \nabla \cdot (h\boldsymbol{u}) = 0, \\ (h\boldsymbol{u})_t + \nabla \cdot \left( h\boldsymbol{u} \otimes \boldsymbol{u} + \frac{1}{2}gh^2 \boldsymbol{I} \right) = 0, \end{cases} \quad x \in \Omega, \quad t > 0, \tag{30}$$

The dataset is obtained from the PDEBench benchmark (Takamoto et al., 2022), which simulates a 2D radial dam-break scenario. The initial water height is defined as a circular bump centered in the domain:

$$h(\boldsymbol{x}, t = 0) \begin{cases} 2.0, & r < ||\boldsymbol{x}||, \\ 1.0, & r \geq ||\boldsymbol{x}||, \end{cases} \tag{31}$$

where the radius $r$ is sampled uniformly from the interval $(0.3, 0.7)$.

- **Schrödinger Equations:**

$$\begin{aligned} \text{Linear:} \quad & i\psi_t + \frac{1}{2}\Delta\psi + V(\boldsymbol{x})\psi = 0, \quad \boldsymbol{x} \in \Omega, \quad t > 0, \\ \text{Nonlinear:} \quad & i\psi_t + \frac{1}{2}\Delta\psi + \lambda||\psi||^2\psi = 0, \quad \boldsymbol{x} \in \Omega, \quad t > 0, \end{aligned} \tag{32}$$

We solve both the linear and nonlinear Schrödinger equations using the Strang splitting method (Strang, 1968) combined with the Fast Fourier Transform (FFT) (Kumar et al., 2019). The initial condition is constructed as:

$$u_0 = \sum_{k=1}^{5} (a_k + b_k i) e^{ikx + \phi_k} \tag{33}$$

where $a_k, b_k \sim \mathcal{N}(0, 1)$ are drawn from a standard normal distribution and $\phi_k \sim \mathcal{U}(0, 2\pi)$ are uniformly sampled phases.

# C. Visualizations for FNO and Our Method

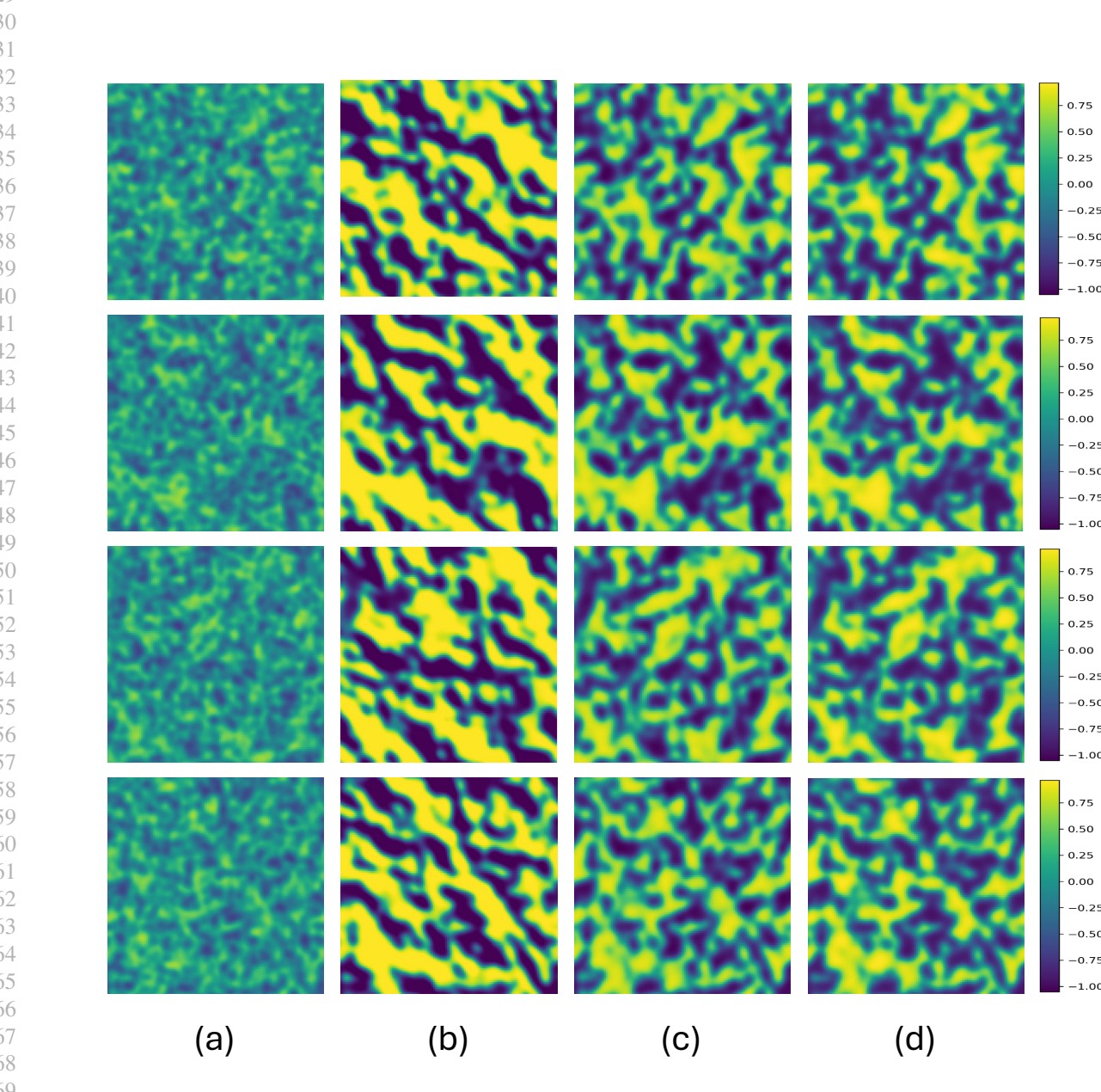

(a)    (b)    (c)    (d)

*Figure 3.* Visualization for the conservative Allen Cahn equation at $T = 2$. (a): initial condition (b): FNO (c): FNO with adaptive correction (d): ground truth

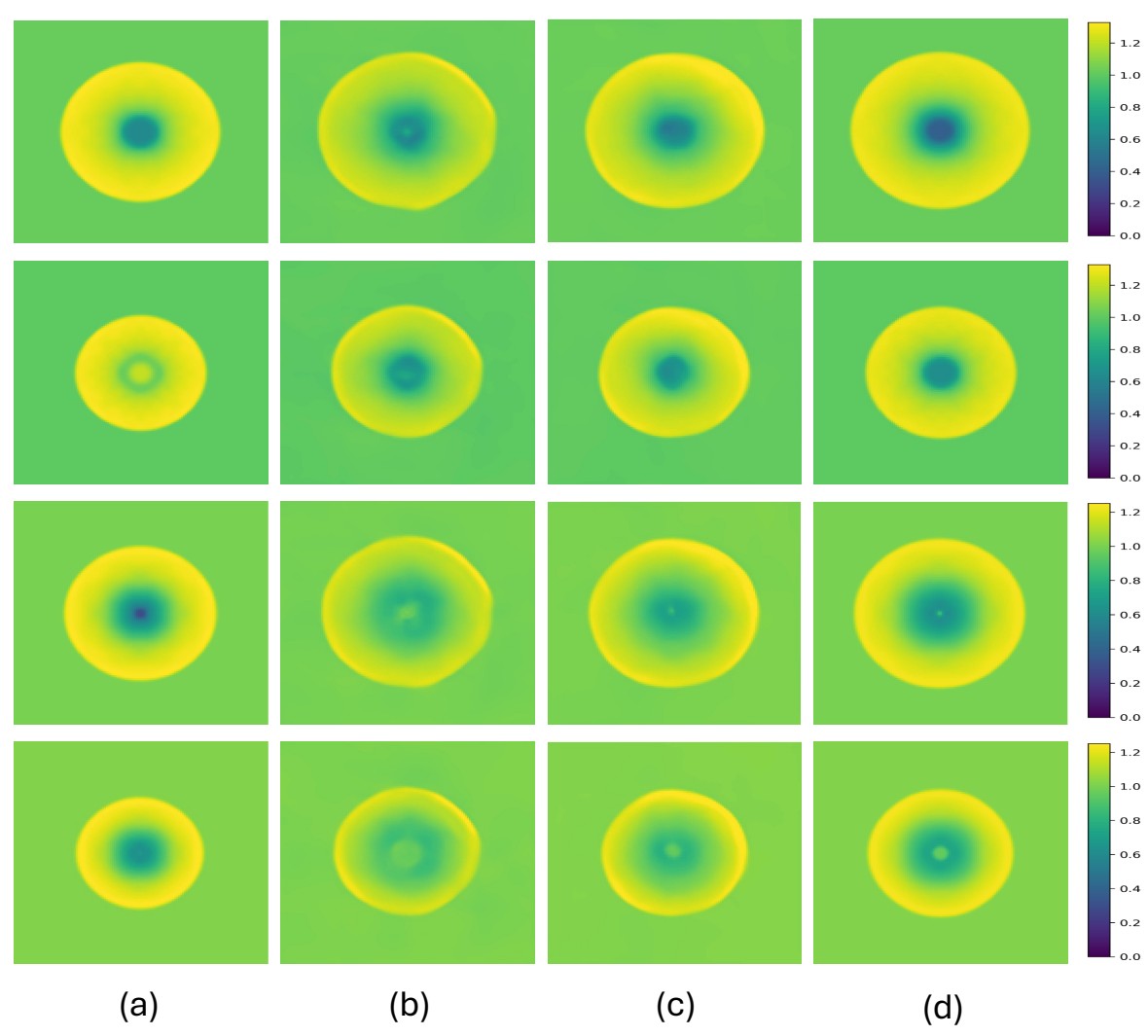

*Figure 4.* Visualization for the shallow water equation at $T = 0.03$. (a) Initial condition; (b) FNO prediction; (c) FNO with adaptive correction; (d) Ground truth.

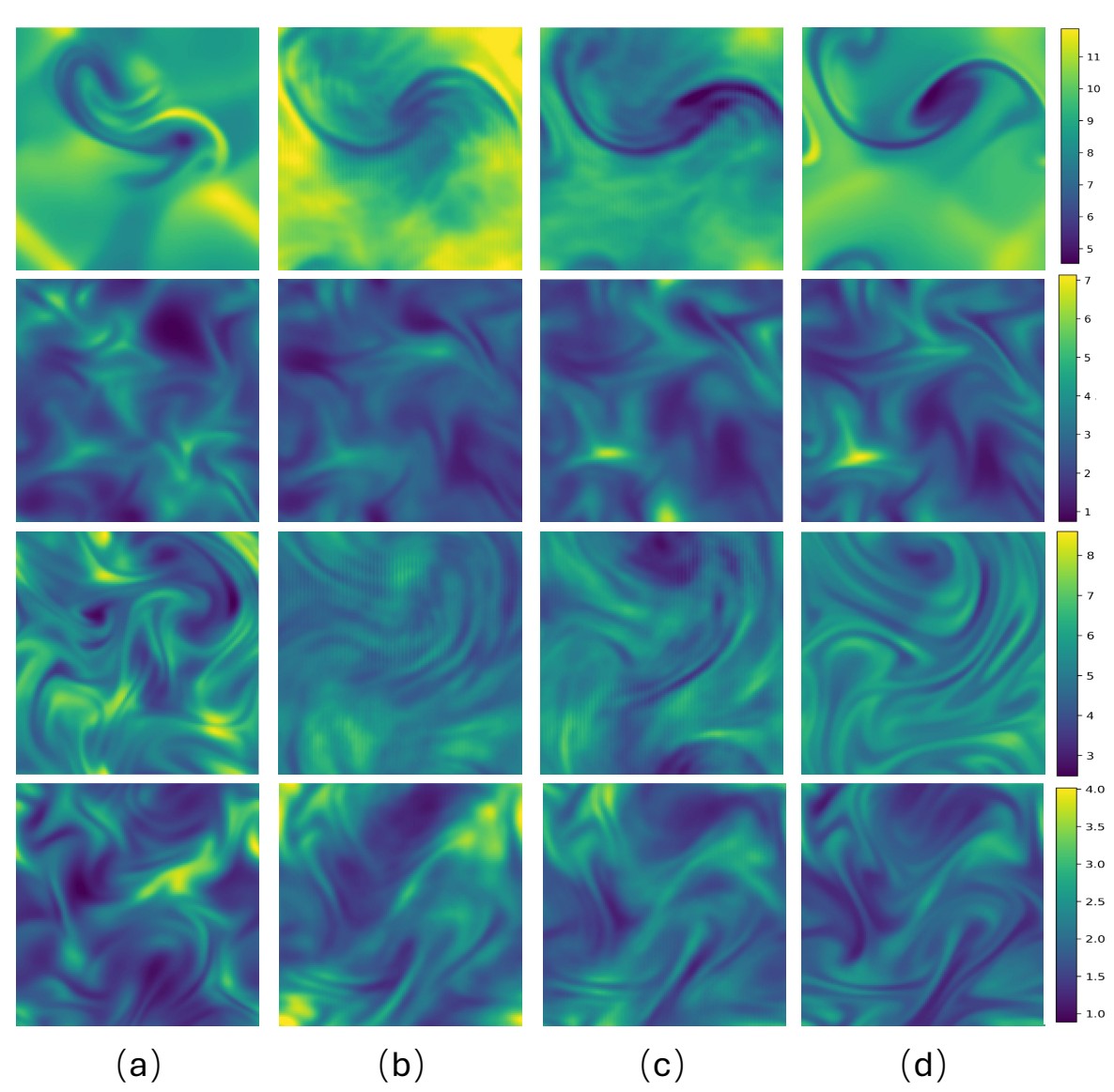

(a)     (b)     (c)     (d)

*Figure 5.* Visualization of the density $\rho$ for compressible Navier-Stokes equation at $T = 0.5$. (a) Initial condition; (b) FNO prediction; (c) FNO with adaptive correction; (d) Ground truth.

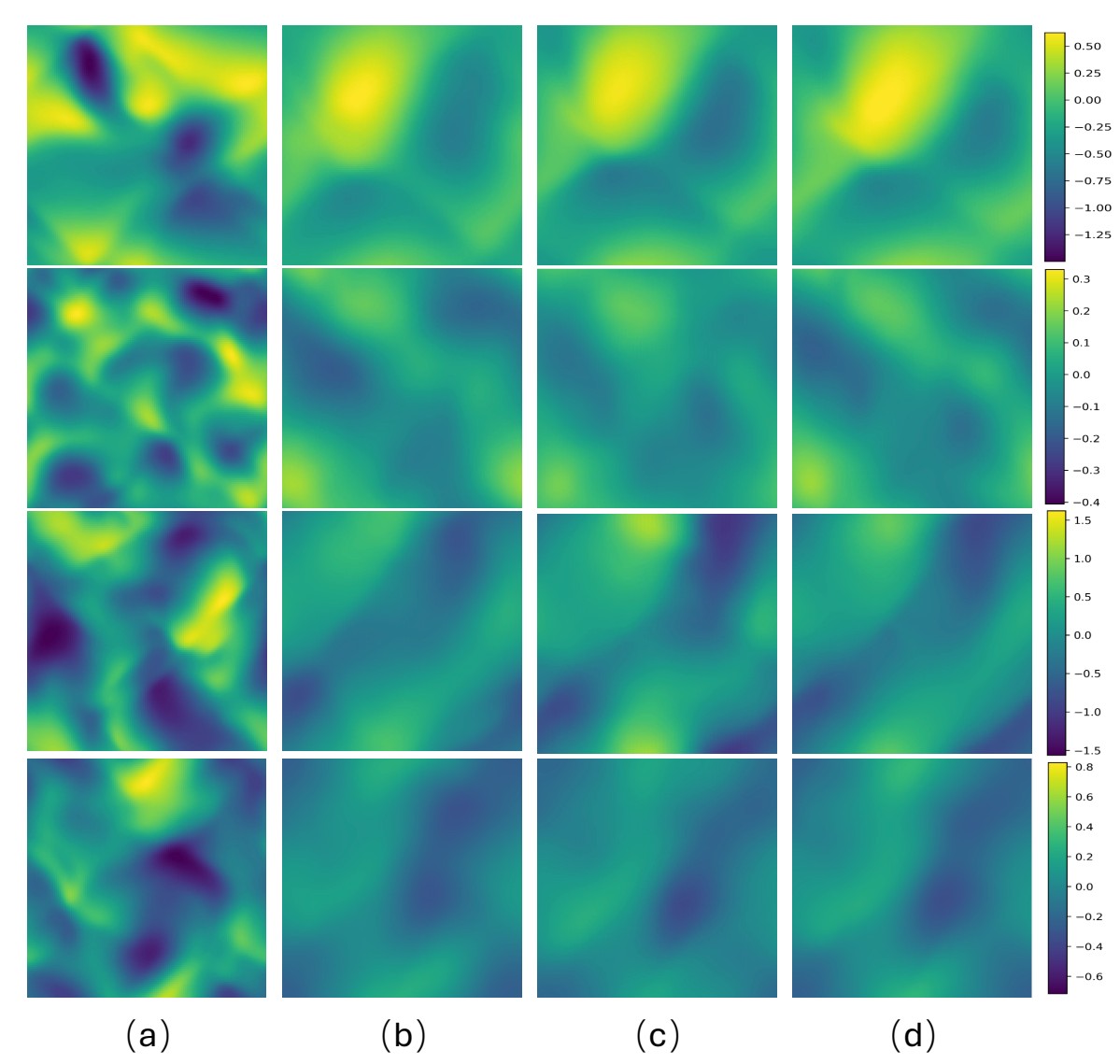

(a)        (b)        (c)        (d)

*Figure 6.* Visualization of the corresponding velocity in the $x$-direction for Figure 5. (a) Initial condition; (b) FNO prediction; (c) FNO with adaptive correction; (d) Ground truth.

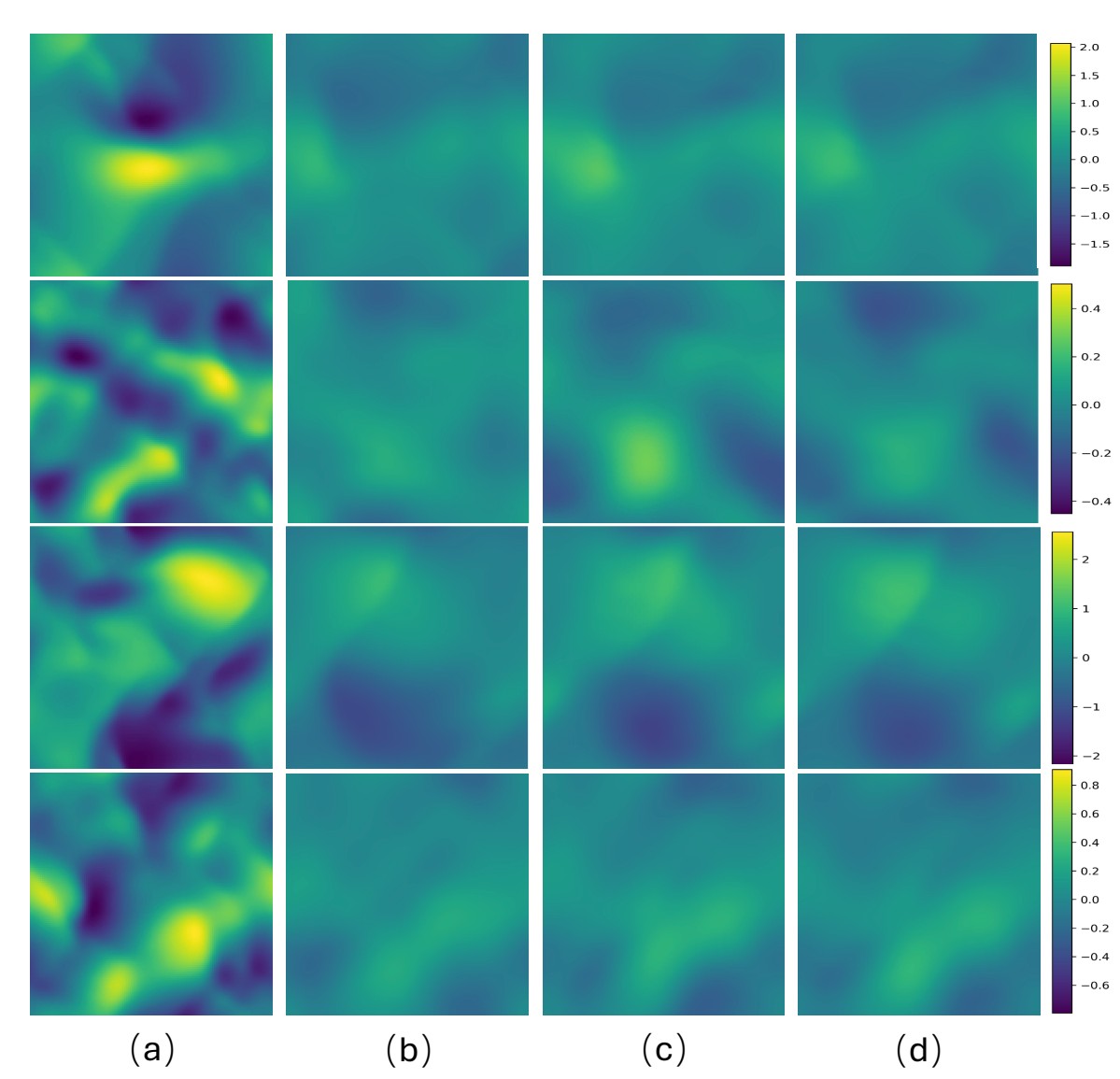

(a)        (b)        (c)        (d)

*Figure 7.* Visualization of the corresponding velocity in the $y$-direction for Figure 5. (a) Initial condition; (b) FNO prediction; (c) FNO with adaptive correction; (d) Ground truth.

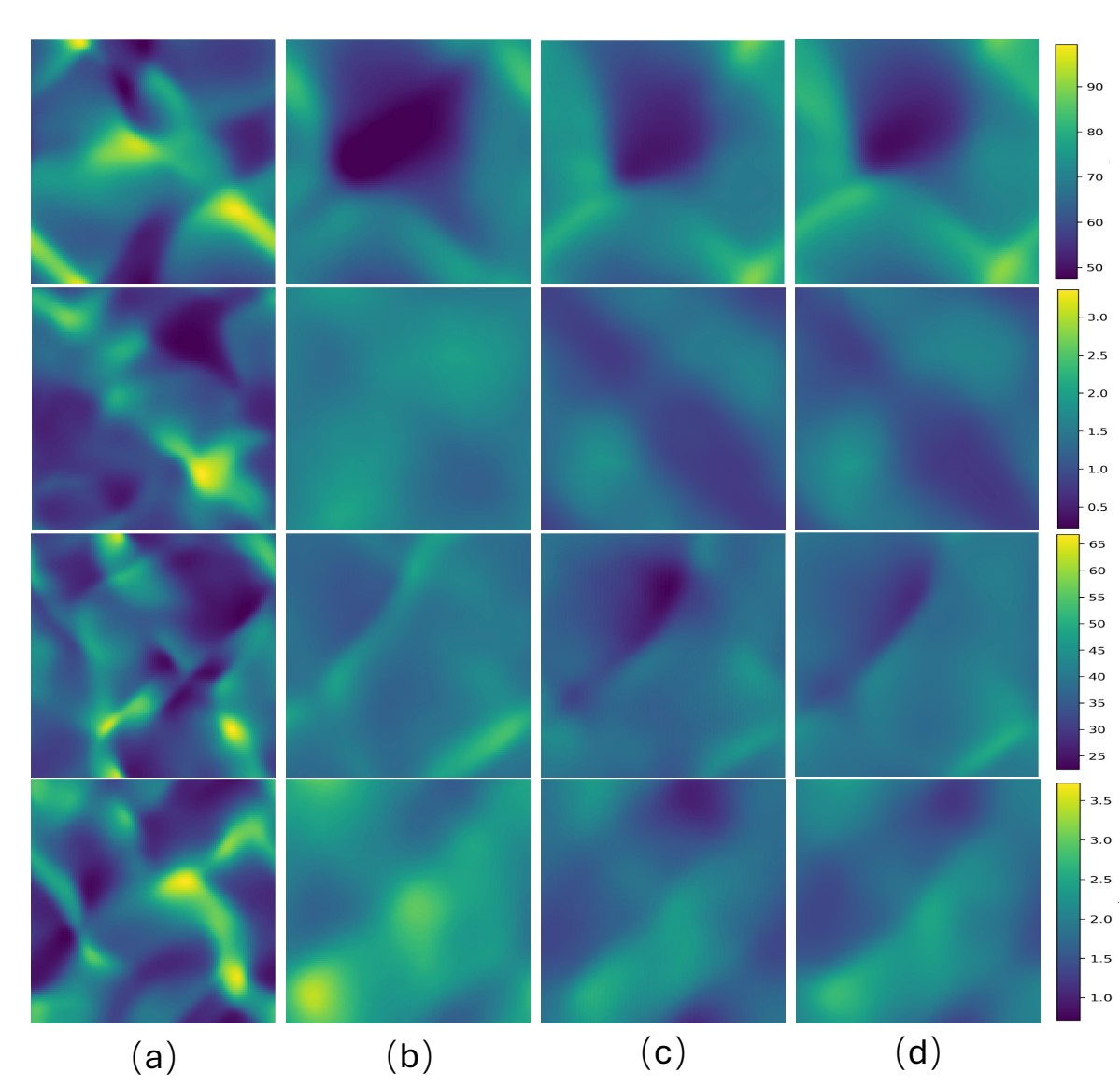

(a)          (b)          (c)          (d)

*Figure 8.* Visualization of the corresponding pressure $p$ for Figure 5. (a) Initial condition; (b) FNO prediction; (c) FNO with adaptive correction; (d) Ground truth.

