# OpenReview forum: "Adaptive Correction for Ensuring Conservation Laws in Neural Operators"
_ICML.cc/2026/Conference — Submitted to ICML 2026_

### Official Review · Reviewer_K1hm · 2026-02-16

**Soundness:** 2
**Presentation:** 2
**Significance:** 2
**Originality:** 3
**Overall Recommendation:** 4
**Confidence:** 4

**Summary:**

This paper proposes an adaptive, learnable correction module for enforcing conservation laws in neural operators during rollout. The authors provide a theoretical guarantee for their methods and conduct experiments on several PDE benchmarks with different backbones.

**Compliance With Llm Reviewing Policy:**

Affirmed.

**Final Justification:**

The authors have addressed most of my concerns. However, according to their 3D experiments, the correlation drops quickly after 3 or 4 steps (below 0.8), so I think the additional 3D experiments in the rebuttal are meaningless. Therefore, my score is "weak accept" for this paper.

**Key Questions For Authors:**

1. How sensitive are the results to larger dt and longer rollout steps?

2. What is the training and inference overhead introduced by the correction module?

3. Can you report results on at least one more recent neural operator architecture?

4. For FNO, does the correction preserve resolution invariance in practice?

**Limitations:**

yes

**Strengths And Weaknesses:**

Strengths

- The authors introduce a learnable correction that enforces hard conservation constraints while maintaining flexibility and expressivity. This idea feels novel to me.

- The proposed method is flexible and plug-and-play across different neural operator backbones.

- The paper provides a theoretical guarantee, and shows that the corrected formulation does not reduce expressivity and is not worse in optimal reconstruction loss under the stated setup.

- The ablation studies are relatively rich and helpful for understanding the effect of the correction module.

Weaknesses

My major concern is that the experimental evaluation is relatively weak in scope and does not fully support the claims about rollout reliability. Including,

a. The experiments are limited to 1D and 2D cases on regular grids.

b. The rollout horizon in the benchmark PDEs is relatively short (e.g., ~4–10 steps), and the time step size is small (e.g., dt ≈ 0.01–0.05), which may reduce the difficulty of long-term rollout stability.

c. The model backbones are relatively old (UNet, GTNO, FNO). The authors should evaluate the module on more recent neural operator models to demonstrate broader relevance.

d. A comprehensive computation cost analysis is required, especially compared with other approaches such as projection methods.

e. A resolution-invariance test for FNO is missing.

---

> ### Author Rebuttal · Authors · 2026-03-30
>
> We sincerely thank the reviewer for the constructive feedback on the experimental evaluation. We have conducted extensive additional experiments to address your concerns, including **3D settings, large time stepsize dt, long rollout stability, additional neural operator architectures, and resolution generalization**. The new results consistently demonstrate the effectiveness, robustness, and general applicability of our method.
>
> 1. **No 3D experiments.**
>
>    **Response.** We have conducted additional experiments on the 3D MHD\_64 dataset from the Well benchmark (NeurIPS 2024) using both UNet and FNO. The results (VRMSE of density) are shown below:
>     | Rollout | UNet| Ours| FNO | Ours |
>    |-------|-----|------|-----|------|
>    | 0:1 | 0.289|**0.265** | 0.275 | **0.257** |
>    |0:20 |2.351|**1.236**|1.311|**1.167**|
>
>    The results show that our method consistently improves accuracy, with particularly significant gains in long rollout (0:20), demonstrating its effectiveness in reducing error accumulation in 3D settings.
>
>
> 2. **How sensitive are the results to larger dt and longer rollout steps?**
>
>    **Response.** We conducted additional experiments on the shallow water equation with varying time stepsizes ($dt$) and rollout lengths. The results are shown below:
>
>     | dt | rollout| FNO| Ours |
>    |-------|-----|------|-----|
>    | 0.05| 0:1|0.35e-2 | **0.32e-2**|
>    |     | 0:5|0.72e-2 | **0.68e-2** |
>    |     | 0:10|1.01e-2 | **0.87e-2** |
>    |     | 0:20|3.45e-2 | **1.71e-2** |
>    | 0.10| 0:1|0.53e-2 | **0.46e-2** |
>    |     | 0:5|1.18e-2 | **1.04e-2** |
>    |     | 0:10|1.44e-2 | **1.18e-2** |
>
>    These results show that our method consistently improves performance across different time steps. More importantly, the improvement becomes more pronounced in long rollout settings, indicating enhanced stability and reduced error accumulation over time.
>
> 3. **What is the training and inference overhead introduced by the correction module?**
>
>    **Response.** The proposed correction module is computationally efficient, as it consists of a lightweight pointwise MLP. The parameter count and training time are reported below:
>
>    | Model | FNO | Ours |
>    |-------|-----|------|
>    | Params | 37836360 | 37837481|
>    | Time/epoch | 4.88s | 4.94s |
>
>    This demonstrates that the proposed method introduces negligible overhead in both model size and training time.
>
>
> 4. **Can the authors report results on at least one more recent neural operator architecture?**
>
>    **Response.** We thank the reviewer for the suggestion. We have conducted additional experiments on Transolver (NeurIPS 2023) and LNO (NeurIPS 2024) for the conservative Allen--Cahn and the shallow water equation. The results are shown below:
>    | Model | Transolver |  Ours(Transolver)  |   LNO   |   Ours(LNO) |
>    |-------|------------|-----------|---------|-----------|
>    | conservative Allen--Cahn |  4.14e-2   | **2.35e-2** | 3.69e-1 | **7.69e-2** |
>    | Shallow Water    |  1.47e-3   | **1.16e-3** | 4.47e-2 | **3.26e-2** |
>
>    These results demonstrate that our method is architecture-agnostic and consistently improves performance across diverse neural operator designs.
>
> 5. **A resolution-invariance test for FNO is missing.**
>
>    **Response.** We thank the reviewer for pointing this out. The correction operator for FNO is implemented via a pointwise MLP, which operates independently at each spatial location and does not depend on the underlying grid resolution. As a result, it preserves the resolution-invariant property of FNO.
>
>     To further validate this, we conducted additional experiments on zero-shot super-resolution for the shallow water equation. The results are shown below:
>     | Resolution | 128 | 256 | 384 | 512 |
>     |------------|-----|-----|-----|-----|
>     | FNO        | 2.57e-3 | 3.01e-3| 3.74e-3 | 3.14e-3 |
>     | Ours       | 2.32e-3| 2.77e-3 | 3.22e-3 | 2.81e-3 |
>
>     The results show consistent improvements across all resolutions, confirming that the proposed correction preserves the resolution generalization capability of FNO.
>
> We hope these additional results and clarifications address the reviewer’s concerns and better highlight the effectiveness and generality of our method. We remain fully open to further discussion if there are any remaining points, and we would appreciate it if the reviewer could re-evaluate our work accordingly.

---

> > ### Author Rebuttal · Reviewer_K1hm · 2026-04-01
> >
> > Thank you for the additional experiments. The new results are helpful, especially for the 3D setting and longer rollouts. However, I still find the current evidence insufficient to assess the significance of the reported improvements.
> >
> > First, for the 3D experiments, please report the step-wise correlation coefficient over the rollout, not only the final VRMSE values. This would help clarify how prediction quality evolves over time.
> >
> > Second, it would be very helpful if the authors could provide an anonymous link for supplementary visualizations, including prediction vs. reference comparisons, error maps, and energy spectrum comparisons. With only a few scalar numbers reported in the table, it is difficult to judge whether the improvements are physically meaningful.
> >
> > Finally, please clarify which conservation laws are considered in this 3D experiment.

---

> > > ### Author Response · Authors · 2026-04-01
> > >
> > > We thank the reviewer for the quick follow-up and for the constructive suggestions. For the 3D MHD dataset, the underlying system satisfies **mass conservation**, which is the constraint enforced in our experiments. We apologize for not making this explicit in the original rebuttal and will clarify this in the revision.
> > >
> > > Following the reviewer’s suggestions, we have provided comprehensive visualizations via the following anonymous link:
> > > https://anonymous.4open.science/r/mhd-comparison-visualizations-EC4F/
> > >
> > > The supplementary materials include:
> > > - **Step-wise Pearson correlation over rollout**;
> > > - **Prediction vs. reference comparisons** and corresponding **error maps** evaluated using both VRMSE and NRMSE (as defined in the Well dataset);
> > > - **Energy spectrum comparisons**.
> > >
> > > All results are reported for both **UNet vs. MC-UNet** and **FNO vs. MC-FNO**, across multiple rollout steps.
> > >
> > > For 3D visualization, we follow standard practice and present **2D slices extracted from the 3D fields**. Specifically, we visualize slices at indices `[0, 31, 63]` along one spatial dimension, which provide representative views of the spatial structure and error distribution.
> > >
> > > We hope these additional results and visualizations provide a clearer picture of the improvements, particularly in terms of temporal stability and physical fidelity, and adequately address the reviewer's concerns.

---

### Official Review · Reviewer_UwBd · 2026-03-05

**Soundness:** 3
**Presentation:** 3
**Significance:** 3
**Originality:** 2
**Overall Recommendation:** 5
**Confidence:** 4

**Summary:**

This paper introduces a method for exactly enforcing a single linear or quadratic conservation law into an operator network. Importantly, the method can be applied on top of any existing operator network architecture. In the case of a linear conservation law, the method is essentially to perform a weighted average of a fixed set of projections onto the constraint, with the weights being learned parameters. In the quadratic case, a more complicated method with more learnable parameters is introduced. Numerical experiments show significant improvements are attained by enforcing conservation laws in this manner.

**Compliance With Llm Reviewing Policy:**

Affirmed.

**Key Questions For Authors:**

in situations which are more complicated than one dimensions, how are the local correction operators chosen? I think it would be important to give more details here. Also, did you perform a study comparing different choices of local correction operators? It is possible that the method is very sensitive to this choice.

On line 239, there is a broken equation reference. Please fix this.

**Limitations:**

yes

**Strengths And Weaknesses:**

Strengths:

- The method which is introduced is simple and easy to apply on top of any existing neural operator with minimal additional training cost.
- The constraints are exactly enforced by the method.
- Numerical experiments demonstrate the efficacy of the proposed method.
- The paper is well-written and easy to follow.

Weaknesses:

- For the linear conservation laws, the method requires fixing a collection of projections onto the constraint set (this corresponds to the local correction operators introduced in Section 3). The output of the method is a weighted average of these projections. Is there any guidance on how to choose this initial set of projections? This seems to me to be a bit ad hoc.
- The proposed methods can only enforce a single conservation law at a time. Although, for linear conservation laws, the methods can easily be modified to enforce multiple linear conservation laws, I believe.

---

> ### Author Rebuttal · Authors · 2026-03-30
>
> We sincerely thank the reviewer for the positive assessment of our work and for the constructive questions. We are glad that the reviewer finds the paper valuable, and we provide clarifications below.
>
> 1. **How are the local correction operators chosen? Is this method sensitive to high dimensions?**
>
>    **Response.** The local correction operators are defined in Eq. (5). Each local operator adjusts the corresponding entry of the output to enforce the conservation constraint, ensuring that the global invariant is satisfied.
>
>     To evaluate scalability, We have conducted additional experiments on the 3D MHD\_64 dataset from the Well benchmark (NeurIPS 2024) using both UNet and FNO. The results (VRMSE of density) are shown below:
>
>    | Rollout | UNet| Ours| FNO | Ours |
>    |-------|-----|------|-----|------|
>    | 0:1 | 0.289|**0.265** | 0.275 | **0.257** |
>    |0:20 |2.351|**1.236**|1.311|**1.167**|
>
>    The results show that our method remains effective in higher-dimensional settings, demonstrating its applicability beyond 2D problems.
>    We agree that exploring different designs of local correction operators is an interesting direction, and we will include this discussion as future work in the revision.
>
> 2. **On line 239, there is a broken equation reference. Please fix this.**
>
>    **Response.** We thank the reviewer for the careful reading. The reference should point to Eq. (20), and we will correct this in the revised version.
>
> We thank the reviewer again for the positive assessment and constructive feedback, and we are glad that the reviewer finds the work valuable.

---

> > ### Author Rebuttal · Reviewer_UwBd · 2026-04-02
> >
> > I thank the authors for their clarifications. I stand by my score of 5.

---

> > > ### Author Response · Authors · 2026-04-03
> > >
> > > We sincerely thank the reviewer for the continued support and for engaging with our work throughout the review process. We really appreciate your time and thoughtful feedback.

---

### Official Review · Reviewer_cqp5 · 2026-03-09

**Soundness:** 3
**Presentation:** 3
**Significance:** 2
**Originality:** 3
**Overall Recommendation:** 4
**Confidence:** 4

**Summary:**

This paper introduces a method for ensuring linear and quadratic conservation laws are strictly followed in neural operator outputs. The proposed method is modular and learnable. The authors demonstrate their method on a range of mass and norm conservation problems, showing strict conservation and improved prediction performance compared to baselines.

**Compliance With Llm Reviewing Policy:**

Affirmed.

**Final Justification:**

I raise my initial score to a 4 since the authors have addressed the issues I raised, which were particularly regarding significance. The new experiments conducted by the authors have convinced me that their method's significance is sufficient.

**Key Questions For Authors:**

1. The authors write integral versions of the linear and quadratic conservation laws in the paper. Does this also satisfy linear and nonlinear differential laws (e.g., various PDE equations themselves as constraints)?
2. How much were the $\lambda$ tuned for physics conservation loss experiments?
3. Do the authors have any intuition as to why the conservation error for CNS is > 1000 even with the physics conservation loss?

**Limitations:**

Yes, the authors discuss limitations in their approach, including its applicability to only linear and quadratic conservation laws and not more general types.

**Strengths And Weaknesses:**

**Strengths:**
The paper is clearly written, and the proposed method is well explained. The authors’ architecture is simple but it accurately addresses the problem of ensuring linear and quadratic conservation laws in operator learning, with minimal overhead while being “plug-and-play.” These are important qualities that many prior works have lacked. This makes the paper’s presentation, originality, and soundness good.

**Weaknesses:**
My main area of improvement with this paper is its significance. While the proposed method works well for linear and quadratic conservation laws, the authors only demonstrate their performance across a few standard benchmark examples but do not extend their method to an application where strict conservation is truly necessary, such as a control problem or a setting where strict conservation enables a significant improvement in prediction accuracy. I recommend the authors look into such applications to strengthen their paper.

Neural operators have the benefit of being agnostic to the input and output resolutions. How does the proposed method extend to zero-shot super-resolution on the target problems?

While the relative L2 error is a good metric, it is only a point estimate which favors blurring and low frequencies. I recommend the authors compare their method against others using other metrics, such as the decay of the Fourier power spectrum, to analyze the performance of their method. For instance, does their method provide a more faithful spectrum decay rate in zero shot super resolution compared to the baseline?

In Remark 3.2, the authors suggest that Theorem 3.1 indicates that their model may train more stably than an architecture using the constraint loss as a soft loss. However, I do not see exactly how this is derived from the theorem. It seems to me that the theorem only refers to behavior at the global minimum but does not discuss the difference in the optimization landscapes between the soft loss and constrained settings.

**Minor issues:**
Undefined reference in the proof of Theorem 3.1.

---

> ### Author Rebuttal · Authors · 2026-03-30
>
> We sincerely thank the reviewer for the detailed and thoughtful feedback. We have carefully considered the comments and provide clarifications and additional experimental results below to address the concerns.
>
> 1. **Significance and Practical Necessity**
>
>    **Response.** We thank the reviewer for this important question. The necessity of strict conservation becomes critical in many real-world systems, where violation of conservation laws leads to physically invalid or unusable solutions, especially in long-term simulations.
>
>    For example, in quantum mechanics, the Schrödinger equation preserves the $L^2$ norm of the wave function, which represents the total probability and must remain exactly equal to 1. Any deviation implies that the total probability is no longer conserved, making the solution physically invalid and unreliable for downstream tasks such as expectation estimation or measurement prediction.
>
>     Similarly, in fluid dynamics, violations of mass or energy conservation can lead to artificial creation or loss of physical quantities. While such errors may appear small initially, they accumulate over time and can dominate the solution in long rollout settings, leading to unstable or nonphysical behavior.
>
>      These issues are not merely quantitative inaccuracies but fundamental violations of physical principles. In contrast to loss-based approaches that only approximately enforce conservation, our method guarantees exact satisfaction of the constraints, ensuring physically consistent solutions throughout the evolution. We will clarify this motivation in the revision.
>
> 2. **Zero-shot Super-resolution Performance**
>
>    **Response.** We have conducted additional experiments on zero-shot super-resolution using FNO for the shallow water equation. The results below show that our method preserves the resolution-invariant property of FNO.
>
>    |Resolution|128|256|384|512|
>    |--|--|--|--|--|
>    |FNO|2.57| 3.01|3.74|3.14|
>    |Ours |2.32 |2.77|3.22|2.81|
>
>     This is because the proposed correction operator is implemented via a pointwise MLP, which operates independently at each spatial location and does not depend on grid resolution. Therefore, it naturally inherits the resolution-invariance of FNO, which also relies on pointwise operations for cross-resolution generalization.
>
> 3. **Why does Theorem 3.1 suggest more stable training compared to soft constraint methods?**
>
>    **Response.** The theorem itself does not directly prove optimization stability. It shows that the constrained optimization problem can be achieved by training a model equipped with the proposed adaptive correction, without explicitly introducing a constraint loss.
>
>     In practice, soft-constraint methods require balancing data fidelity and constraint satisfaction, which often leads to sensitivity to the choice of $\lambda$ and competing gradients during training. In contrast, our method enforces conservation by construction, eliminating this trade-off and leading to more stable and robust optimization behavior, as also supported by our empirical results.
>
> 4. **Do the conservation laws satisfy corresponding differential laws?**
>
>    **Response.** Yes. Conservation laws are closely related to the underlying governing differential equations. For example, mass conservation is expressed in differential form by the continuity equation $$\frac{\partial \rho}{\partial t} + \nabla \cdot (\rho \mathbf{v}) = 0.$$ More generally, conservation laws are the global (integral) form of PDEs describing local dynamics. Our method enforces these global invariants at the solution level, complementing the local modeling performed by neural operators.
>
> 5. **Hyperparameter Tuning ($\lambda$) for Soft Constraints**
>
>    **Response.** We carefully tune $\lambda$ over a wide range of values, and the tested values are reported in Table 5.
>
>    We note that this sensitivity to $\lambda$ reflects an inherent limitation of loss-based approaches, where constraint enforcement depends on balancing competing objectives.
>
> 6. **Why is the conservation error for CNS extremely large (>1000) with the loss-based method?**
>
>    **Response.** The large conservation error in the CNS case is primarily due to two factors: (1) the overall prediction error is significantly larger than for other equations, which amplifies violations of conservation constraints; and (2) the physical quantities (e.g., density) have a much larger dynamic range (approximately 0--15 compared to 0--1.5 in other tasks), further magnifying the absolute conservation error.
>
>    This highlights a key limitation of loss-based approaches: the effectiveness of constraint enforcement depends on both prediction accuracy and scale. In contrast, our method enforces conservation exactly, independent of these factors.
>
> We hope that the established practical necessity, clarifications and additional experimental results resolve the reviewer's concerns and hope the reviewer considers a positive re-evaluation of our work.

---

> > ### Author Rebuttal · Reviewer_cqp5 · 2026-04-03
> >
> > I thank the authors for their responses; they have addressed some of my points, including super-resolution experiments and some of my questions. However, they have not addressed a couple important points as far as I can see: (1) expanding the experimental section with practical applications and (2) alternative metrics other than relative L2.
> >
> > For this reason, I have decided to maintain my score for now.

---

> > > ### Author Response · Authors · 2026-04-06
> > >
> > > We sincerely thank the reviewer for the careful follow-up and for pointing out the remaining concerns. We apologize for not fully addressing these points in our previous response. Below, we provide additional experiments and clarifications regarding (1) practical applications and (2) alternative evaluation metrics.
> > >
> > > ---
> > >
> > > **(1) Practical applications**
> > >
> > > While the effect of conservation may be less apparent in single-step prediction, it becomes increasingly important in **long-rollout settings**, which are critical in practical simulations. This behavior is already observed in Fig. 2 and Appendix C of our manuscript. To further demonstrate this, we conducted additional long-rollout experiments on the shallow water equation (with 1-step $dt=0.05$):
> > >
> > > | rollout | 0:1 | 0:10 | 0:20 |
> > > |--------|------|------|------|
> > > | FNO    | 0.35e-2 | 1.01e-2 | 3.45e-2 |
> > > | Ours   | **0.32e-2** | **0.87e-2** | **1.71e-2** |
> > > | Improvement | 8.6% | 13.9% | 50.4% |
> > >
> > > The results show that the improvement becomes substantial as the rollout length increases, highlighting the importance of enforcing conservation for stable long-term prediction.
> > >
> > > In addition, we conducted experiments on **3D magnetohydrodynamic (MHD) simulations**, which provide a representative physical application where strict conservation is essential. MHD systems are governed by the continuity equation, enforcing **mass conservation** throughout the evolution. In such systems, density is tightly coupled with momentum, pressure, and magnetic field dynamics. As a result, even small violations of mass conservation can accumulate over time, leading to drift in global quantities and degradation of the solution.
> > >
> > > We evaluate both UNet and FNO with and without our method:
> > >
> > > | Rollout | UNet | Ours | FNO | Ours |
> > > |--------|------|------|------|------|
> > > | 0:1    | 0.289 | **0.265** | 0.275 | **0.257** |
> > > | 0:20   | 2.351 | **1.236** | 1.311 | **1.167** |
> > >
> > > Our method consistently improves accuracy, with particularly significant gains in long rollouts (0:20), demonstrating both its effectiveness and practical importance in MHD systems.
> > >
> > > ---
> > >
> > > **(2) Alternative metrics**
> > >
> > > To address the concern regarding evaluation metrics beyond relative $L^2$ error, we compute the **Fourier power spectrum error** (measured as RMSE in log-space) and **Pearson Correlation** for the 3D MHD simulations. The mean error over rollout 0:10 is shown below:
> > >
> > > |Metrics| UNet | Ours | FNO | Ours |
> > > |------|------|------|------|------|
> > > |Fourier power spectrum| 0.86 | **0.71** | 0.48 | **0.30** |
> > > |Pearson Correlation| 0.07 | **0.18** | 0.17 | **0.27** |
> > >
> > > These results show that our method maintains higher correlation with the reference solution and better preserves spectral properties.
> > >
> > > We observe an inherent decay in correlation across all methods as errors accumulate during long-term rollout. In this light, the consistently higher correlation sustained by our approach demonstrates its superior temporal robustness and physical fidelity.
> > >
> > > We also provide additional visualizations to complement these metrics:
> > >
> > > - **Fourier power spectrum decay visualizations** (see [UNet](https://anonymous.4open.science/r/mhd-comparison-visualizations-EC4F/viz_mhd_unet_mcunet/spectrum_step10.png) and [FNO](https://anonymous.4open.science/r/mhd-comparison-visualizations-EC4F/viz_mhd_fno_mcfno/spectrum_step10.png)), illustrating improved preservation of spectral structure;
> > > - **Step-wise Pearson correlation over rollout** (see [UNet](https://anonymous.4open.science/r/mhd-comparison-visualizations-EC4F/viz_mhd_unet_mcunet/correlation_vs_rollout.png) and [FNO](https://anonymous.4open.science/r/mhd-comparison-visualizations-EC4F/viz_mhd_fno_mcfno/correlation_vs_rollout.png)), showing more stable temporal behavior.
> > >
> > > ---
> > >
> > > We hope these additional experiments and analyses help address the reviewer’s concerns regarding both practical significance and evaluation metrics. We would greatly appreciate it if the reviewer could take these into account in the final assessment.

---

### Official Review · Reviewer_XRX6 · 2026-03-10

**Soundness:** 2
**Presentation:** 3
**Significance:** 2
**Originality:** 4
**Overall Recommendation:** 5
**Confidence:** 4

**Summary:**

The paper "Adaptive Correction for Ensuring Conservation Laws in Neural Operators" proposes an adaptive correction module that can be attached to a neural operator to make its outputs exactly satisfy conservation laws, without hard-coding the architecture or relying on fragile penalty terms. Instead of a fixed projection step, it learns a lightweight correction (a learnable vector/operator) that adjusts predictions to satisfy the constraint, while preserving model expressivity. The authors test it on several PDE benchmarks (transport, conservative Allen–Cahn, shallow water, compressible Navier–Stokes, linear/nonlinear Schrödinger), and report improved accuracy and long-rollout stability compared to (i) loss-penalty conservation, and (ii) projection-based enforcement.

**Compliance With Llm Reviewing Policy:**

Affirmed.

**Final Justification:**

The authors addressed all our concerns satisfactorily. We are happy to raise the score to 5.

**Key Questions For Authors:**

1.) page 2: the authors discuss that their approach is learnable and adaptive i.e. the correction operator is directly learnt from data. I was wondering how does data quality affect this? What if the data itself breaks conservation laws due to noisy data? How will this affect the method? This is nowhere discussed.

2.) for the linear conservation law presented on page 3 the authors state this can be a single physical quantity such as density or a product of density and velocity. I was wondering if this is then only referring to total velocity or if it can also refer to component wise products, i.e. can the u be vector valued?

3.) there is a broken reference on page 5

4.) Experiments: the authors mention a UNet as a type of operator architecture. To my knowledge a UNet is not the classical type of operator architecture, is it?

5.) Why were only 2D experiments presented and no 3D examples? It would strengthen the results if also 3D examples were shown / tested.

6.) page 7: Can the fundamental loss-based methods maybe improved by using better optimisers to deal with competing loss gradients form the different objectives? how would that then compare to the results presented here?

7.) I can't really see / judge the sensitivity to the Lambda parameter from table 5. Can this be improved?

**Limitations:**

yes

**Strengths And Weaknesses:**

Soundness: The theoretical part is solid and well-motivated. The numerical tests, however, are rather simple, in particular: (i) the numerical and analytical solutions of the proposed equations are rather smooth, thus usually easy to learn for a neural network-based model, and (ii) only in one case is the reduction of error propagation presented (which is perhaps the most relevant motivation for this work).

Presentation: The presentation of the work is very clear and of a high technical level, both with regards to the theoretical part and the numerical experiments.

Significance: The problem addressed is extremely relevant, particularly in light of the ever-increasing use of neural operators as substitutes for numerical methods. However, the improvements presented compared to the baseline are modest, probably due to the relative simplicity of the solutions to be approximated. Additionally, the work is limited to a subset of conservation laws i.e. linear and quadratic conservation laws.

Originality: To the best of my knowledge, the work is original.

---

> ### Author Rebuttal · Authors · 2026-03-30
>
> We sincerely thank the reviewer for the comprehensive and constructive feedback, and for recognizing the originality and solid theoretical foundation of our work. We have carefully addressed all concerns and conducted additional experiments (including noisy data and 3D settings) to further validate our method. Detailed responses are provided below.
>
> 1. **How does data quality affect the correction operator?**
>
>    **Response.** We thank the reviewer for this insightful question. To investigate robustness to imperfect data, we conducted additional experiments where small Gaussian noise is added to both the inputs and ground-truth outputs. The results are shown below:
>     | $\sigma$ | rollout| FNO| Ours |
>    |--|--|--|--|
>    | 0.005| 0:1|2.89e-3 | **2.61e-3**|
>    || 0:20|1.33e-2 | **9.55e-3** |
>    | 0.010| 0:1|2.99e-3 | **2.83e-3**|
>    || 0:20|1.33e-2| **1.01e-2**|
>
>    The results show that our method consistently outperforms the baseline FNO under noisy conditions, with more pronounced improvements in long rollout settings. The proposed correction enforces exact conservation at the solution level, constraining the model outputs to remain within the physically valid manifold. This reduces the impact of noise and leads to more stable predictions over time.
>
> 2. **Can $u$ be vector-valued?**
>
>    **Response.** Yes, the formulation naturally extends to vector-valued quantities. In such cases, the conserved quantity (e.g., momentum $\rho v$) can be vector-valued, and the correction operator is applied accordingly. The learnable coefficient $A$ becomes vector-valued as well, enabling adaptive correction for each component. We will clarify this point in the paper.
>
> 3. **Why do the authors mention a UNet as a type of operator architecture?**
>
>    **Response.** We thank the reviewer for pointing this out. Strictly speaking, UNet is not a neural operator in the classical sense. In this work, we include UNet as a representative baseline that is widely used in operator learning benchmarks (e.g., PDEBench (Neurips22) and The Well (Neurips24)).
>    Our goal is to demonstrate that the proposed correction module is **architecture-agnostic** and can be seamlessly integrated into both specialized neural operators (e.g., FNO, GTNO) and standard convolutional architectures such as UNet.
>
> 4. **There is a broken reference on page 5.**
>
>    **Response.** We thank the reviewer for the careful reading. The reference should point to Eq.~(20), and this will be corrected in the revised version.
>
> 5. **It would strengthen the results if also 3D examples were shown / tested.**
>
>    **Response.** We appreciate this constructive suggestion. We have added additional experiments on a 3D problem (MHD\_64) using the Well dataset (NeurIPS 2024), evaluated on both UNet and FNO. The results (VRMSE of density) are shown below:
>     | Rollout | UNet| Ours| FNO | Ours |
>    |--|--|--|--|--|
>    | 0:1 | 0.289|**0.265** | 0.275 | **0.257** |
>    |0:20 |2.351|**1.236**|1.311|**1.167**|
>
>    The results show that our method consistently improves performance in 3D settings. Notably, the gains are more significant in long rollout scenarios, demonstrating its effectiveness in reducing error accumulation. This indicates that the proposed method scales well to higher-dimensional problems.
>
> 6. **Can loss-based methods be improved with better optimizers?**
>
>    **Response.** We agree that improved optimization strategies may partially alleviate training difficulties. However, they cannot fundamentally resolve the limitation of loss-based approaches. The core issue lies in the need to balance competing objectives (data fidelity vs. constraint satisfaction), which is inherently sensitive to the choice of $\lambda$ (the weight in front of the conservation penalty) and often leads to unstable optimization dynamics.
>    In contrast, our method enforces conservation **by construction**, eliminating this trade-off entirely and ensuring exact satisfaction of the constraint without additional tuning.
>
> 7. **How does Table 5 show the sensitivity to the $\lambda$ parameter?**
>
>    **Response.** Table 5 evaluates loss-based methods across a range of $\lambda$ values. While the sensitivity is moderate for mass conservation, it becomes significantly more pronounced in the case of norm conservation. For example, increasing $\lambda$ from $10^{-4}$ to $10^{-3}$ leads to a sharp degradation in accuracy (from 8.17% to 90.1%), indicating instability.
>    This highlights a fundamental limitation of loss-based approaches: constraint enforcement depends critically on the choice of $\lambda$, and even small changes can lead to drastically different outcomes. In contrast, our method enforces conservation exactly and does not require such tuning.
>
> We hope these additional results and clarifications address the reviewer’s concerns and will be reflected in the final evaluation.

---

> > ### Author Rebuttal · Reviewer_XRX6 · 2026-04-01
> >
> > We thank the authors for their answers to our comments. All our concerns are addressed. We are happy to raise our score to 5.

---

> > > ### Author Response · Authors · 2026-04-01
> > >
> > > We sincerely thank the reviewer for the positive feedback and for taking the time to re-evaluate our work. We are very glad that our responses and additional experiments have addressed the concerns. We truly appreciate your support.

---

### Decision · Program_Chairs · 2026-04-30

**Decision:**

Reject

**Comment:**

This paper's topic is significant as most reviewers said. This tackles the problem of learning conservative laws. They propose a special training method for this purpose, focusing on linear and quadratic cases. After that, they choose three base models and several standard PDEs and show that in most cases, those base models are enhanced with their proposed training mechanism.

Overall, reviewers are positive and I agree on its significance. But, there are a couple of points to pay attention. First, experiment PDEs are all standard basic PDEs (as one reviewer said). Second, it lacks comparisons with recent methods. They need to show the efficacy in recent methods. Since recent methods are able to learn better than FNO and other base models. So it is unclear whether this method is beneficial to the state-of-the-art models. Third, real-world settings may hinder adopting this method since exact PDEs are frequently missing (or unknown) only with observed quantities.